

# Modelling thermomechanical ice deformation using a GPU-based implicit pseudo-transient method (FastICE v1.0)

Ludovic Räss[1], Aleksandar Licul[2,3], Frederic Herman[2,3], Yury Y. Podladchikov[3,4], and Jenny Suckale[1]

[1]Stanford University, Geophysics Department, 397 Panama Mall, Stanford CA 94305, USA.
[2]Institute of Earth Surface Dynamics, University of Lausanne, 1015 Lausanne, Switzerland.
[3]Swiss Geocomputing Centre, University of Lausanne, 1015 Lausanne, Switzerland.
[4]Institute of Earth Sciences, University of Lausanne, 1015 Lausanne, Switzerland.

**Correspondence:** Ludovic Räss (ludovic.rass@gmail.com)

**Abstract.** Accurate predictions of future sea level rise require numerical models that capture the complex thermomechanical feedbacks in rapidly deforming ice. Shear margins, grounding zones and the basal sliding interface are locations of particular interest where the stress-field is complex and fundamentally three-dimensional. These transition zones are prone to thermomechanical localisation, which can be captured numerically only with high temporal and spatial resolution. Thus, better under-
standing the coupled physical processes that govern these boundaries of localised strain necessitates a non-linear, full Stokes model that affords high resolution and scales well in three dimensions. This paper's goal is to contribute to the growing toolbox for modelling thermomechanical deformation in ice by levering GPU accelerators' parallel scalability. We propose a numerical model that relies on pseudo-transient iterations to solve the implicit thermomechanical coupling between ice motion and temperature involving shear-heating and a temperature-dependant ice viscosity. Our method is based on the finite-difference
discretisation, and we implement the pseudo-time integration in a matrix-free way. We benchmark the mechanical Stokes solver against the finite-element code Elmer/Ice and report good agreement among the results. We showcase a parallel version of the solver to run on GPU-accelerated distributed memory machines, reaching a parallel efficiency of 93%. We show that our model is particularly useful for improving our process-based understanding of flow localisation in the complex transition zones bounding rapidly moving ice.

## 1  Introduction

The fourth IPCC report (Solomon et al., 2007) revealed that existing ice sheet flow models do not accurately describe polar ice sheet discharge (e.g., Gagliardini et al., 2013; Pattyn et al., 2008) owing to their inability to simultaneously model slow and fast ice flow (Gagliardini et al., 2013; Bueler and Brown, 2009). This issue results from the fact that many ice flow models are based on simplified approximations of non-linear Stokes equations, such as shallow ice models (Bueler and Brown, 2009; Bassis,
2010; Schoof and Hindmarsh, 2010; Goldberg, 2011; Egholm et al., 2011; Pollard and DeConto, 2012; Perego et al., 2012; Tezaur et al., 2015). Shallow ice models are computationally more tractable and describe the motion of large homogeneous portions of ice as a function of the basal friction. However, this category of models fails to capture the coupled multi-scale processes that govern the behaviour of the boundaries of streaming ice, including shear margins, grounding zones and the





basal interface. These boundaries dictate the stability of the current main drainage routes from Antarctica and Greenland, and

predicting their future evolution is critical for understanding polar ice sheet discharge.

Full Stokes models (Gagliardini and Zwinger, 2008; Gagliardini et al., 2013; Jarosch, 2008; Jouvet et al., 2008; Larour et al., 2012; Leng et al., 2012, 2014; Brinkerhoff and Johnson, 2013; Isaac et al., 2015) provide a complete mechanical description of deformation by capturing the entire stress-rate and strain-rate tensor. In three dimensions (3-D), full Stokes calculations set a high demand on computational resources that requires a parallel and high-performance computing approach to achieve

reasonable times to solution. An added challenge in full Stokes models is ice's strongly non-linear thermomechanics. Ice's viscosity significantly depends on both temperature and strain-rate (Robin, 1955; Hutter, 1983; Morland, 1984), which can lead to spontaneous localisation of shear (e.g., Duretz et al., 2019; Räss et al., 2019a). Particularly challenging is the scale separation associated with localisation, which leads to micro-scale physical interaction generating meso-scale features such as thermally-activated shear zones or preferential flow paths in macro-scale ice domains. Thus, both high spatial and temporal

resolutions are important for numerical models to capture and resolve spontaneous localisation.

This paper's main contribution is to lever the unprecedented parallel performance of modern graphical processing units (GPUs) to accelerate the time-to-solution for thermomechanically coupled full Stokes models in 3-D utilising a pseudo-transient (PT) iterative scheme – FastICE (Räss et al., 2019). We argue that our numerical model is particularly useful for advancing our process-based understanding of the boundaries of streaming flow including shear margins, grounding zones and

the basal sliding interface. We demonstrate our thermomechanical Stokes models' ability to resolve the spontaneous ice flow localisation in both 2-D and 3-D and on (multiple) GPUs.

Recent trends in the computing industry show a shift from single-core to many-core architectures as an effective way to increase computational performance. This trend is common to both central processing unit (CPU) and GPU hardware architectures (Cook, 2012). The programming model behind GPUs is based on a parallel principle called Single Instruction Multiple

Data (SIMD). This principle entails that every single instruction is executed on different data. The same instructions block is executed by every thread. GPUs' massive parallelism and the related high performance is achieved by executing thousands of threads concurrently using multi-threading in order to effectively hide latency. Numerical stencil-based techniques such as the finite-difference method allow one to take advantage of GPU hardware, since spatial derivatives are approximated by differences between two (or more) adjacent grid-points. This results in minimal, local and regular memory access patterns. The

operations performed on each stencil are identical for each grid-point throughout the entire computational domain. Combined with a matrix-free discretisation of the equations and iterative PT updates, the finite-difference stencil evaluation is well suited for the SIMD programming philosophy of GPUs. Each operation on the GPU assigns one thread to compute the update of a given grid-point. Since on the GPU device, one core can simultaneously execute several threads, the operation set is executed on the entire computational domain almost concurrently.

We tailor our numerical method to optimally exploit the massive parallelism of GPU hardware (Omlin, 2017; Räss et al., 2018; Duretz et al., 2019; Räss et al., 2019a). Our numerical implementation relies on an iterative and matrix-free method to solve the mechanical and thermal problems using a finite-difference discretisation on a Cartesian staggered grid. We ensure optimal performance, minimising the memory footprint bottleneck while ensuring optimal data alignment in computer memory.





Our accelerated PT algorithm (Frankel, 1950; Cundall et al., 1993; Poliakov et al., 1993; Kelley and Keyes, 1998; Kelley and Liao, 2013) utilises an analogy of transient physics to converge to the steady-state problem at every time step. One advantage of this approach is that the iterative stability criterion is physically motivated and intuitive to adjust and to generalise. Using transient physics for numerical purpose allows us to define local CFL-like criteria in each computational cell to be used to minimise residuals. This approach enables maximal convergence rate simultaneously in the entire domain and avoids costly global reduction operations from becoming a bottleneck in parallel computing.

We verify the numerical implementation of our mechanical Stokes solver against available benchmark studies including EISMINT (Huybrechts and Payne, 1996) and ISMIP (Pattyn et al., 2008). There is only one model inter-comparison that investigates the coupled thermomechanical dynamics, EISMINT 2 (Payne et al., 2000). Unfortunately, experiments in EISMINT 2 are usually performed using a coupled thermomechanical first-order shallow ice model (Payne and Baldwin, 2000; Saito et al., 2006; Hindmarsh, 2006; Bueler et al., 2007; Hindmarsh, 2009; Brinkerhoff and Johnson, 2015) making the comparison to our full Stokes implementation less immediate. Although thermomechanically coupled Stokes models exist (Zwinger et al., 2007; Leng et al., 2014; Schäfer et al., 2014; Gilbert et al., 2014; Zhang et al., 2015; Gong et al., 2018), very few studies have investigated key aspects of the implemented model, such as convergence among grid refinement and impacts of one-way vs. two-way couplings, with few exceptions (e.g. Duretz et al., 2019).

We start by providing an overview over the mathematical model, describing ice dynamics and its numerical implementation. We then discuss GPUs capabilities and explain our GPU implementation. We further report model comparison against a selection of benchmark studies, followed by sharing the results and performance measurements. Finally, we discuss pros and cons of the method, and highlight glaciological contexts in which our model could prove useful. The codes examples based on the PT method in both MATLAB and CUDA C programming language are available for download from Bitbucket at https://bitbucket.org/lraess/ice/ and from http://wp.unil.ch/geocomputing/software/ice.

# 2 The model

## 2.1 The mathematical model

We capture the flow of an incompressible, non-linear, viscous fluid – including a temperature-dependent rheology. Since ice is approximately incompressible, the equation for conservation of mass reduces to:

$$\frac{\partial v_i}{\partial x_i} = 0 \,, \tag{1}$$

where $v_i$ is the velocity component in the spatial direction $x_i$.

Neglecting inertial forces, ice's flow is driven by gravity and is resisted by internal deformation and basal stress:

$$\frac{\partial \tau_{ij}}{\partial x_j} - \frac{\partial P}{\partial x_i} + F_i = 0 \,, \tag{2}$$





where $F_i = \rho g \sin(\alpha)[1, 0, -\cot(\alpha)]$ is the external force. Ice density is denoted by $\rho$, $g$ is the gravitational acceleration, and $\alpha$ is the characteristic bed slope. $P$ is the isotropic pressure and $\tau_{ij}$ is the deviatoric stress tensor. The deviatoric stress tensor $\tau_{ij}$ is obtained by decomposing the Cauchy stress tensor $\sigma_{ij}$ in terms of deviatoric stress $\tau_{ij}$ and isotropic pressure $P$.

In the absence of phase transitions, the temporal evolution of temperature in deforming, incompressible ice is governed by advection, diffusion and shear-heating:

$$\rho c \left( \frac{\partial T}{\partial t} + v_i \frac{\partial T}{\partial x_i} \right) = \frac{\partial}{\partial x_i} \left( k \frac{\partial T}{\partial x_i} \right) + \tau_{ij} \dot{\epsilon}_{ij} \, , \tag{3}$$

where $T$ represents the temperature deviation from the initial temperature $T_0$, $c$ is the specific heat capacity, $k$ is the spatially-varying thermal conductivity and $\dot{\epsilon}_{ij}$ is the strain-rate tensor. The term $\tau_{ij} \dot{\epsilon}_{ij}$ represents the shear-heating, a source term that emerges from the mechanical model.

Shear-heating could locally raise the temperature in the ice to the pressure melting point. Once ice has reached melting point, any additional heating is converted to latent heat, which prevents further temperature increase. Thus, we impose a temperature cap at the pressure melting point, following Suckale et al. (2014), by describing the melt production using a heavy-side function $\theta(T - T_m)$:

$$\rho c \left( \frac{\partial T}{\partial t} + v_i \frac{\partial T}{\partial x_i} \right) =$$
$$\frac{\partial}{\partial x_i} \left( k \frac{\partial T}{\partial x_i} \right) + [1 - \theta(T - T_m)] \tau_{ij} \dot{\epsilon}_{ij} \, , \tag{4}$$

where $T_m$ stands for the ice melting temperature. We balance the heat produced by shear-heating with a sink term in regions where the melting temperature is reached. The volume of produced meltwater can be calculated in a similar way as proposed by Suckale et al. (2014).

We approximate the rheology of ice through Glen's flow law (Glen, 1952; Nye, 1953):

$$\dot{\epsilon}_{ij} = \frac{1}{2} \left( \frac{\partial v_i}{\partial x_j} + \frac{\partial v_j}{\partial x_i} \right)$$
$$= a_0 \tau_{II}^{n-1} \exp\left( -\frac{Q}{R(T + T_0)} \tau_{ij} \right) \, , \tag{5}$$

where $a_0$ is the pre-exponential factor, $R$ is the universal gas constant, $Q$ is the activation energy, $n$ is the stress exponent, and $\tau_{II}$ is the second invariant of the stress tensor defined by $\tau_{II} = \sqrt{1/2 \tau_{ij} \tau_{ij}}$. Glen's flow law posits an exponent of $n = 3$.

At the ice top surface $\Gamma_t(t)$, we impose the upper surface boundary condition $\sigma_{ij} n_j = -P_{atm} n_j$, where $n_j$ denotes the normal unit vector at the ice surface boundary, and $P_{atm}$ the atmospheric pressure. Because atmospheric pressure is negligible relative to pressure within ice column, we can also use a standard stress-free simplification of the upper surface boundary condition $\sigma_{ij} n_j = 0$. On the bottom ice-bedrock interface, we can impose two different boundary conditions. For the parts of the ice-bedrock interface $\Gamma_0(t)$ where the ice is frozen to the ground, we impose a zero velocity $v_i = 0$ and thus no sliding boundary condition. On the parts of ice-bedrock interface $\Gamma_s(t)$ where the ice is at the melting point, we impose a Rayleigh





friction boundary condition – the so-called linear sliding law – given by:

$$v_i n_i = 0 \, ,$$

$$n_i \sigma_{ij} t_j = -\beta^2 v_j t_j \, ,$$
(6)

where the parameter $\beta^2$ denotes a given sliding coefficient, $n_i$ denotes the normal unit vector at the ice-bedrock interface, and $t_j$ denotes any unit vector tangential to the bottom surface. On the side or lateral boundaries, we impose either Dirichlet boundary conditions if the velocities are known, or periodic boundary conditions, mimicking an infinitely extended domain.

## 2.2 Non-dimensionalisation

For numerical purposes and for ease of generalisation, it is often preferable to use non-dimensional variables. This allows one to limit truncation errors (especially relevant for single-precision calculations) and to scale the results to various different initial configurations. Here, we use two different scale sets, depending on whether we solve the purely mechanical part of the model or the thermomechanically coupled system of equations.

In the case of an isothermal model, we use ice thickness, $H$, and gravitational driving stress to non-dimensionalise the governing equations:

$$\overline{L} = H \, ,$$

$$\overline{\tau} = \rho g \overline{L} \sin(\alpha), \, ,$$
(7)

$$\overline{v} = 2^n A_0 \overline{L} \overline{\tau}^n \, ,$$

where $A_0$ is the isothermal deformation rate factor and $\alpha$ is the mean bed slope. We can then rewrite the governing equations in their non-dimensional form as follows:

$$\frac{\partial v_i'}{\partial x_i'} = 0 \, ,$$

$$\frac{\partial \tau_{ij}'}{\partial x_j'} - \frac{\partial P'}{\partial x_i'} + F_i' = 0 \, ,$$
(8)

$$\dot{\epsilon}'_{ij} = \frac{1}{2} \left( \frac{\partial v_i'}{\partial x_j'} + \frac{\partial v_j'}{\partial x_i'} \right) = 2^{-n} \tau_{\mathrm{II}}'^{\,n-1} \tau_{ij}' \, ,$$

where $F_i'$ is now defined as $F_i' = [1, 0, -\cot(\alpha)]$. The model parameters are the mean bed slope $\alpha$ and domain size in each horizontal direction, i.e. $\mathrm{L}_x'$ and $\mathrm{L}_y'$.

Reducing the thermomechanically coupled equations to a non-dimensional form requires not only length and stress, but also temperature and time. We choose the characteristic scales such that the coefficients in front of the diffusion and shear-heating



terms in the temperature evolution Eq. (3) reduce to one:

$$\overline{T} = \frac{nRT_0{}^2}{Q} \,,$$

$$\overline{\tau} = \rho c_p \overline{T} \,,$$

$$\overline{t} = 2^{-n} a_0^{-1} \overline{\tau}^{-n} \exp\left(\frac{Q}{RT_0}\right) \,, \qquad (9)$$

$$\overline{L} = \sqrt{\frac{k}{\rho c_p} \overline{t}} \,.$$

These choices entail that the velocity scale in the thermomechanical model is $\overline{v} = \overline{L}/\overline{t}$. We obtain the non-dimensional (primed-variables) by using the characteristic scales given in Eq. (9), which leads to:

$$\frac{\partial v_i'}{\partial x_i'} = 0 \,,$$

$$\frac{\partial \tau_{ij}'}{\partial x_j'} - \frac{\partial P'}{\partial x_i'} + F_i' = 0 \,,$$

$$\frac{\partial T'}{\partial t'} + v_i' \frac{\partial T'}{\partial x_i'} = \frac{\partial^2 T'}{\partial x_i'^2} + \tau_{ij}' \dot{\epsilon}_{ij}' \,, \qquad (10)$$

$$\dot{\epsilon}_{ij}' = \frac{1}{2}\left(\frac{\partial v_i'}{\partial x_j'} + \frac{\partial v_j'}{\partial x_i'}\right)$$

$$= 2^{-n} \tau_{\mathrm{II}}'^{\,n-1} \exp\left(\frac{nT'}{1 + \frac{T'}{T_0'}}\right) \tau_{ij}' \,,$$

where $F_i'$ is now defined as $F_i' = \overline{F}\,[1, 0, -\cot(\alpha)]$ and $\overline{F} = \rho g \sin(\alpha) \overline{L}/\overline{\tau}$. The model parameters are the non-dimensional initial temperature $T_0'$, the stress exponent $n$, the non-dimensional force $\overline{F}$, the mean bed slope $\alpha$, non-dimensional domain height $\mathrm{L}_z'$, and the horizontal domain size $\mathrm{L}_x'$ and $\mathrm{L}_y'$ (Figure 3). We motivate the chosen characteristic scales by their usage in other studies of thermomechanical strain localisation (Duretz et al., 2019; Kiss et al., 2019). In the interest of a simple notation, we will omit the prime symbols on all non-dimensional variables in the remainder of the paper.

## 2.3  A simplified 1-D semi-analytical solution

We consider a specific 1-D mathematical case where all horizontal derivatives vanish ($\partial/\partial x = \partial/\partial y = 0$). The only remaining shear stress component $\tau_{xz}$ and pressure $P$ are determined by analytical integration and are constant in time considering a fixed domain (Figure 3). We assume that stresses vanish at the surface and we set both horizontal and vertical basal velocity components to 0. We then integrate the 1-D mechanical equation in the vertical direction and substitute it into the temperature





equation, which leads to:

$$\frac{\partial T(z,t)}{\partial t} = \frac{\partial^2 T(z,t)}{\partial z^2} + 2^{(1-n)} \left(\overline{F}\mathrm{L}_z\right)^{(n+1)}$$

$$\left(1 - \frac{z}{\mathrm{L}_z}\right)^{(n+1)} \exp\left(\frac{nT(z,t)}{1 + \frac{T(z,t)}{T_0}}\right),$$

$$v_x(z,t) = 2^{(1-n)} \left(\overline{F}\mathrm{L}_z\right)^n \int\limits_0^z \left(1 - \frac{z}{\mathrm{L}_z}\right)^n \tag{11}$$

$$\exp\left(\frac{nT(z,t)}{1 + \frac{T(z,t)}{T_0}}\right) dz \, .$$

Notably, the velocity and shear-heating terms (Eq. 11) are now a function only of temperature and, thus, of depth and time. To obtain a solution of the coupled system, one only needs to numerically solve for the temperature evolution profile, while the velocity can then be obtained diagnostically by a simple numerical integration.

## 2.4   The numerical implementation

We discretise the coupled thermomechanical Stokes equations (Eq. 10) using the finite-difference method on a staggered Cartesian grid. Among many numerical methods currently used to solve partial differential equations, the finite-difference method is commonly used and has been successfully applied in solving a similar equations' set relating to geophysical problems in geodynamics (Harlow and Welch, 1965; Ogawa et al., 1991; Gerya, 2009). The staggering of the grid provides second-order

accuracy of the method (Virieux, 1986; Patankar, 1980; Gerya and Yuen, 2003; McKee et al., 2008), avoids oscillatory pressure modes (Shin and Strikwerda, 1997), and produces simple yet highly compact stencils. The different physical variables are located at different locations on the staggered grid. Pressure nodes and normal components of the strain-rate tensor are located at the cell centres. Velocity components are located at the cell mid-faces (Figure 1), while shear stress components are located at the cell vertices in 2-D (e.g., Harlow and Welch, 1965). The resulting algorithms are well suited for taking advantage of

modern many-core parallel accelerators, such as graphical processing units (GPUs) (Omlin, 2017; Räss et al., 2018; Duretz et al., 2019; Räss et al., 2019a). Efficient parallel solvers utilising modern hardware provide a viable solution to resolve the computationally challenging coupled thermomechanical full Stokes calculations in 3-D. The power law viscous ice rheology (Eq. 5) exhibits a non-linear dependence on both the temperature and the strain-rate:

$$\eta = \dot{\epsilon}_{\mathrm{II}}^{\frac{1-n}{n}} \exp\left(-\frac{T}{1 + \frac{T}{T_0}}\right), \tag{12}$$

where $\dot{\epsilon}_{\mathrm{II}}$ is the square root of the second invariant of the strain-rate tensor $\dot{\epsilon}_{\mathrm{II}} = \sqrt{1/2\dot{\epsilon}_{ij}\dot{\epsilon}_{ij}}$. We regularise the strain-rate and temperature dependant viscosity $\eta$ to prevent non-physical values for negligible strain-rates, $\eta_{\mathrm{reg}} = 1/(\eta^{-1} + \eta_0^{-1})$. We use a harmonic mean to obtain a naturally smooth transition to background viscosity values at negligible strain-rate $\eta_0$.





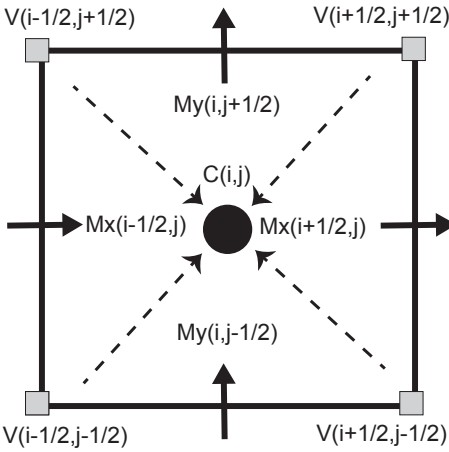

**Figure 1.** Setup of the staggered grid in 2-D. Variable C is located at the cell centre, V depicts variables located at cell vertices and Mx and My represents variables located at cell mid-faces in $x$ or $y$ direction.

We define temperature on the cell centres within our staggered grid. We discretise the temperature equation's advection term using a first-order upwind scheme, doing the physical time integration using either an implicit backward Euler or a Crank-Nicolson (Crank and Nicolson, 1947) scheme.

We rely on a pseudo-transient (PT) continuation or relaxation method to solve the system of coupled non-linear partial differential equations (10) in an iterative and matrix-free way (Frankel, 1950; Cundall et al., 1993; Poliakov et al., 1993; Kelley and Keyes, 1998; Kelley and Liao, 2013). To this end, we reformulate the thermomechanical Eq. (10) in a residual form:

$$-\frac{\partial v_i}{\partial x_i} = f_{\mathrm{p}} \, ,$$

$$\frac{\partial \tau_{ij}}{\partial x_j} - \frac{\partial P}{\partial x_i} + F_i = f_{\mathrm{v}_i} \, ,$$

$$-\frac{\partial T}{\partial t} - v_i \frac{\partial T}{\partial x_i} + \frac{\partial^2 T}{\partial x_i{}^2} + \tau_{ij}\dot\epsilon_{ij} = f_{\mathrm{T}} \, , \tag{13}$$

The right-hand-side terms $(f_{\mathrm{p}}, f_{\mathrm{v}_i}, f_{\mathrm{T}})$ are the non-linear continuity, momentum and temperature residuals, respectively, and quantify the magnitude of the imbalance of the corresponding equations.

We augment the steady-state equations with PT terms using the analogy of physical transient processes such as the bulk compressibility or the inertial terms within the momentum equations (Duretz et al., 2019). This formulation enables us to integrate the equation forward in pseudo-time $\tau$ until we reach the steady-state (i.e. the pseudo-time derivatives vanish). Relying on transient physics within the iterative process provides well-defined (maximal) iterative time step limiters. We reformulate





Eq. (10):

$$-\frac{\partial v_i}{\partial x_i} = \frac{\partial P}{\partial \tau_{\mathrm{p}}} \; ,$$

$$\frac{\partial \tau_{ij}}{\partial x_j} - \frac{\partial P}{\partial x_i} + F_i = \frac{\partial v_i}{\partial \tau_{\mathrm{v}_i}} \; ,$$

$$-\frac{\partial T}{\partial t} - v_i \frac{\partial T}{\partial x_i} + \frac{\partial^2 T}{\partial x_i{}^2} + \tau_{ij}\dot{\epsilon}_{ij} = \frac{\partial T}{\partial \tau_{\mathrm{T}}} \; ,$$

(14)

where we introduced the pseudo-time derivatives $\partial/\partial\tau$ for the continuity ($\partial P/\partial \tau_{\mathrm{p}}$), the momentum ($\partial v_i/\partial \tau_{\mathrm{v}_i}$), and the temperature ($\partial T/\partial \tau_{\mathrm{T}}$) equation.

For every non-linear iteration $k$, we update the effective viscosity $\eta_{\mathrm{eff}}{}^{[k]}$ in the logarithmic space by taking a fraction $\theta_\eta$ of the actual physical viscosity $\eta^{[k]}$ using the current strain-rate and temperature solutions fields and a fraction $(1-\theta_\eta)$ of the effective viscosity calculated in the previous iteration $\eta_{\mathrm{eff}}{}^{[k-1]}$.

$$\eta_{\mathrm{eff}}{}^{[k]} = \exp\left[\theta_\eta \ln\left(\eta^{[k]}\right) + (1-\theta_\eta)\ln\left(\eta_{\mathrm{eff}}{}^{[k-1]}\right)\right] \; ,$$

(15)

where $\theta_\eta$ ($0 \le \theta_\eta \le 1$) is a viscosity relaxation factor. This relaxation of the non-linearity allows the effective viscosity to
iteratively approach its physical value within the pseudo-transient iterations. A similar non-linear viscosity relaxation approach was successfully implemented in the ice sheet model development by Tezaur et al. (2015).

The pseudo-time integration of Eq. (14) leads to the definition of pseudo-time steps $\Delta\tau_{\mathrm{p}}, \Delta\tau_{\mathrm{v}_i}$ and $\Delta\tau_{\mathrm{T}}$, for the continuity, momentum and temperature equations, respectively. Transient physical processes such as compressibility (continuity equation) or acceleration (momentum equation) dictate the maximal allowed explicit pseudo-time step to be utilised in the transient
process. Using the largest stable steps allows one to minimise the iteration count required to reach the steady-state:

$$\Delta\tau_{\mathrm{p}} = \frac{2.1 n_{\mathrm{dim}} \eta_{\mathrm{eff}}^k (1+\eta_{\mathrm{b}})}{\max(n_i)} \; ,$$

$$\Delta\tau_{\mathrm{v}_i} = \frac{\min(\Delta x_i)^2}{2.1 n_{\mathrm{dim}} \eta_{\mathrm{eff}}^k (1+\eta_{\mathrm{b}})} \; ,$$

$$\Delta\tau_{\mathrm{T}} = \left(\frac{2.1 n_{\mathrm{dim}}}{\min(\Delta x_i)^2} + \frac{1}{\Delta t}\right)^{-1} \; ,$$

(16)

where $n_{\mathrm{dim}}$ is the number of dimensions, $\Delta x_i$ and $n_i$ are the grid spacing and the number of grid-points in the $i$ direction ($i = x$ in 1-D, $x, z$ in 2-D and $x, y, z$ in 3-D), respectively. The physical time step, $\Delta t$, advances the temperature in time. The pseudo-time step $\Delta\tau_{\mathrm{T}}$ is an explicit Courant-Friedrich-Lewy (CFL) time step that combines temperature advection and
diffusion. Similarly, $\Delta\tau_{\mathrm{v}_i}$ is the explicit CFL time step for viscous flow, representing the diffusion of strain-rates with viscosity as the diffusion coefficient. It is modified to account for the numerical equivalent of a bulk viscosity $\eta_{\mathrm{b}}$. We choose $\Delta\tau_{\mathrm{p}}$ to be the inverse of $\Delta\tau_{\mathrm{v}_i}$ to ensure that the pressure update is proportional to the effective viscosity, while the velocity update is sensitive to the inverse of the viscosity. This interdependence reduces the iterative method's sensitivity to the variations in the ice's viscosity.





During the iterative procedure, we allow for finite compressibility in the ice, $\partial P/\partial \tau_{\mathrm{p}}$, while assuring that the PT iterations eventually reach the incompressible solution. The relaxation of the incompressibility constraint is analogous to the penalisation of pressure pioneered by Chorin (1967, 1968), and built on extensively subsequently. Compared to projection-type methods, it has the advantage that no pressure boundary condition is necessary that will lead to numerical boundary layers (Weinan and Liu, 1995). We use the parameter $\eta_{\mathrm{b}}$ to balance the divergence-free formulation of strain-rates in the normal stress component

evaluation, where it is multiplied with the pressure residual $f_{\mathrm{p}}$. Thus, normal stress is given by $\tau_{ii} = 2\eta(\dot{\epsilon}_{ii} + \eta_{\mathrm{b}} f_{\mathrm{p}})$. With convergence of the method, the pressure residual $f_{\mathrm{p}}$ vanishes and the incompressible form of the normal stresses is recovered.

Combining the residual notation introduced in Eq. (13), with the pseudo-time derivatives in Eq. (14) leading to the update rules:

$$P^{[k]} = P^{[k-1]} + \Delta P^{[k]} \,,$$

$$v_i^{[k]} = v_i^{[k-1]} + \Delta v_i^{[k]} \,, \tag{17}$$

$$T^{[k]} = T^{[k-1]} + \Delta T^{[k]} \,,$$

where the pressure, velocity and temperature iterative increments represent the current residual $[k]$ multiplied by the pseudo-time step:

$$\Delta P^{[k]} = \Delta \tau_{\mathrm{p}} f_{\mathrm{p}}^{[k]} \,,$$

$$\Delta v_i^{[k]} = \Delta \tau_{\mathrm{v}_i} f_{\mathrm{v}_i}^{[k]} \,, \tag{18}$$

$$\Delta T^{[k]} = \Delta \tau_{\mathrm{T}} f_{\mathrm{T}}^{[k]} \,.$$

The straight-forward update rule (Eq. 17) is based on a first-order scheme ($\partial/\partial \tau$). In 1-D, it implies that one needs $N^2$ iterations to converge to the stationary solution, where $N$ stands for the total number of grid-points. This behaviour arises

because the time step limiter $\Delta \tau_{\mathrm{v}_i}$ implies a second-order dependence on the spatial derivatives for the strain-rates. In contrast, a second-order scheme (Frankel, 1950), $\left(\partial^2/\partial \tau^2 + \partial/\partial \tau\right)$ invokes a wave-like transient physical process for the iterations. The main advantage is the scaling of the limiter as $\Delta x$ instead of $\Delta x^2$ in the explicit pseudo-transient time step definition. We can reformulate the velocity update as:

$$\Delta v_i^{[k]} = \Delta \tau_{\mathrm{v}_i} f_{\mathrm{v}_i}^{[k]} + \left(1 - \frac{\nu}{n_i}\right) \Delta v_i^{[k-1]} \tag{19}$$

where $\alpha$ can be expanded to $(1 - \nu/n_i)$ and acts like a damping term on the momentum residual. A similar damping approach is used for elastic rheology in the FLAC (Cundall et al., 1993) geotechnical software in order to significantly reduce the number of iterations needed for the algorithm to converge. The optimal value of the introduced parameter $\nu$ is found to be in a range $(1 \leq \nu \leq 10)$, and it is usually problem-dependent. This approach was successfully implemented in recent PT developments by Räss et al. (2018, 2019a) and Duretz et al. (2019).

Notably, the PT solution procedure leads to a two-way numerical coupling between temperature and deformation (mechanics), which enables us to recover an implicit solution of the entire system of non-linear partial differential equations. Besides





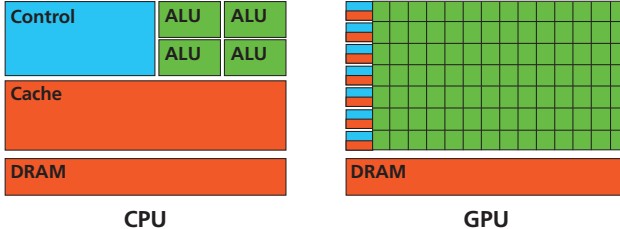

**Figure 2.** Schematic chip representation for both the central processing unit (CPU) and graphical processing unit (GPU) architecture. The GPU architecture consist of thousands of arithmetic and logical units (ALU). On the CPU, most of the on-chip space is devoted to controlling units and cache memory, while the number of ALUs is significantly reduced.

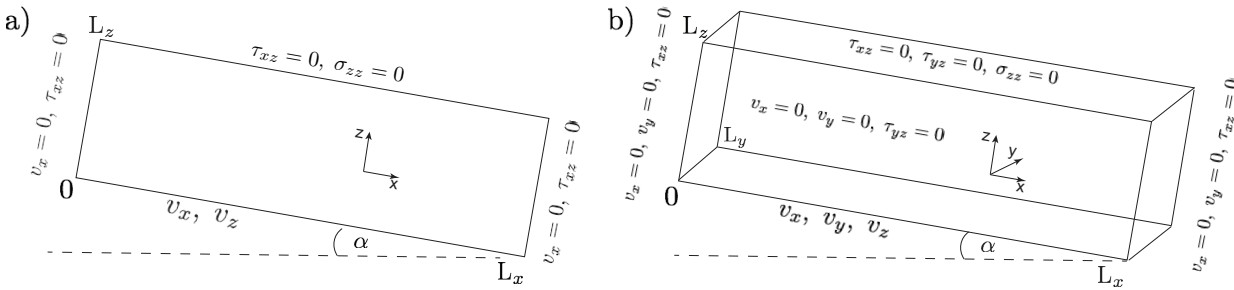

**Figure 3.** Model configuration for the numerical experiments: a) 2-D model and b) 3-D model. Both surface and bed topography are flat but inclined at a constant angle of $\alpha$. We show both the model coordinate axes and the prescribed boundary conditions.

the coupling terms, rheology is also treated implicitly, i.e. the shear viscosity $\eta$ is always evaluated using the current physical temperature, $T$, and strain-rate, $\dot{\epsilon}_{\mathrm{II}}$. Our method is fully local. At no point during the iterative procedure does one need to perform a global reduction, nor to access values that are not directly collocated. These considerations are crucial when designing

a solution strategy that targets parallel hardware such as many-core GPU accelerators. We implemented the PT method in the MATLAB and CUDA C programming languages. Computations in CUDA C can be performed in both double and single precision arithmetic. The computations in CUDA C shown in the remainder of the paper were performed using double-precision arithmetic.

## 3    Levering hardware accelerators

### 3.1    Implementation on graphical processing units

Our GPU algorithm development effort is motivated by the aim to resolve the coupled thermomechanical system of equations (Eq. 12-13) with high spatial and temporal accuracy in 3-D. To this end, we exploit the low-level intrinsic parallelism of shared





memory devices, targeting particularly GPUs. A GPU is a massively parallel device originally devoted to render the colour values for pixels on a screen independently from one another where the latency can be masked by high throughput (i.e. compute

as many jobs as possible in a reasonable time). A schematic representation (Figure 2) highlights the conceptual discrepancy between GPU and CPU. On the GPU chip, most of the area is devoted to the arithmetic units, while on the CPU, a large area of the chip hosts scheduling and control microsystems.

The development of GPU-based solvers requires that one devote time to the design of new algorithms that lever the massively parallel potential of the current GPU architectures. Considerations such as limiting the memory transfers to the mandatory

minimum, avoiding complex data layouts, preferring matrix-free solvers with low memory footprint, and optimal parallel scalability instead of classical Direct-Iterative solver types (Räss et al., 2019a) are key in order to achieve optimal performance.

## 3.2  Multi-GPU implementation

We rely on a distributed memory parallelisation using the message passing interface (MPI) library to overcome the on-device memory limitation inherent to modern GPUs and exploit supercomputers' computing power. Access to a large number of par-

allel processes enables us to tackle larger computational domains or to refine grid resolution. We rely on domain decomposition to split our global computational domain into local domains, each executing on a single GPU handled by an MPI process. Each local process has its boundary conditions defined by a) physics if on the global boundary or b) exchanged information from the neighbouring process in case of internal boundaries. We use CUDA-aware non-blocking MPI messages to exchange the internal boundaries among neighbouring processes. CUDA-awareness allows us to bypass explicit buffer copies on the host

memory by directly exchanging GPU pointers resulting in an enhanced workflow pipe-lining. Our algorithm implementation and solver requires no global reduction. Thus, there is no need for global MPI communication, eliminating an important potential scaling bottleneck. Although the proposed iterative and matrix-free solver features a high locality and should scale by construction, the growing number of MPI processes may deprecate the parallel runtime performance by about 20% owing to the increasing number of messages and overall machine occupancy (Räss et al., 2019b). We address this limitation by overlap-

ping MPI communication and the computation of the inner points of the local domains using streams, a native CUDA feature. CUDA streams allow one to assign asynchronous kernel execution and thus enable the overlap between communication and computation, resulting in optimal parallel efficiency.

## 4  The model configuration

To verify the numerical implementation of the developed PT solver, we consider three numerical experiments based on a

box inclined at a mean slope angle of $\alpha$. We perform these numerical experiments on both 2-D and 3-D computational domains (Figure 3a and 3b, respectively). The non-dimensional computational domains are $\Omega_{2D} = [0 \quad L_x] \times [0 \quad L_z]$ and $\Omega_{3D} = [0 \quad L_x] \times [0 \quad L_y] \times [0 \quad L_z]$ for 2-D and 3-D domains, respectively. The difference between the 2-D and the 3-D configurations lies in the boundary conditions imposed at the base and at the lateral sides. At the surface, the zero stress $\sigma_{ij} n_j = 0$





| Experiment | | $L_x$ | $L_y$ | $\alpha$ | $n$ | $\beta_0$ | $L_x^D$ | $L_y^D$ | $L_z^D$ |
|---|---|---|---|---|---|---|---|---|---|
| Exp. 1 | 2-D | 10 | – | 10 | 3 | – | 2 km | – | 200 m |
| Exp. 1 | 3-D | 10 | 4 | 10 | 3 | – | 2 km | 800 m | 200 m |
| Exp. 2 | 2-D | 10 | – | 0.1 | 3 | 0.1942 | 10 km | – | 1 km |
| Exp. 2 | 3-D | 10 | 10 | 0.1 | 3 | 0.1942 | 10 km | 10 km | 1 km |

**Table 1.** Experiments 1 and 2: Non-dimensional model parameters and the dimensional values $\left(^D\right)$ for comparison.

| Experiment | | $L_x$ | $L_y$ | $L_z$ | $\alpha$ | $n$ | $\overline{F}$ | $T_0$ | $L_x^D$ | $L_y^D$ | $L_z^D$ | $T_0^D$ |
|---|---|---|---|---|---|---|---|---|---|---|---|---|
| Exp. 3 | 1-D | – | – | $3 \times 10^5$ | 10 | 3 | $2.8 \times 10^{-8}$ | 9.15 | – | – | 300 m | -10 °C |
| Exp. 3 | 2-D | $10L_z$ | – | $3 \times 10^5$ | 10 | 3 | $2.8 \times 10^{-8}$ | 9.15 | 3 km | – | 300 m | -10 °C |
| Exp. 3 | 3-D | $10L_z$ | $4L_z$ | $3 \times 10^5$ | 10 | 3 | $2.8 \times 10^{-8}$ | 9.15 | 3 km | 1.2 km | 300 m | -10 °C |

**Table 2.** Experiment 3: Non-dimensional model parameters and the dimensional values $\left(^D\right)$ for comparison

boundary condition is prescribed in all experiments. Experiment 2's model configuration corresponds to the ISMIP benchmark

(Pattyn et al., 2008), where experiment C relates to the 3-D case and experiment D relates to the 2-D case.

Experiments 1 and 2 seek to first verify the implementation of the mechanical part of the Stokes solver, which is the computationally most expensive part (Eq. 8). For these experiments, we assume that the ice is isothermal and neglect temperature. We compare our numerical solutions to the solutions obtained by the commonly used finite-element Stokes solver Elmer/Ice (Gagliardini et al., 2013), which has been thoroughly tested (Pattyn et al., 2008; Gagliardini and Zwinger, 2008). Experiment

3 is a thermomechanically coupled case. The model parameters are the stress exponent $n$, the mean bed slope $\alpha$ and the two horizontal distances $L_x$ and $L_y$ in their respective dimensions $(x, y)$, and appear in Table 1. If a linear basal sliding law (Eq. 6) is prescribed, the respective 2-D and 3-D sliding coefficients are:

$$
\begin{aligned}
\beta^2(x) &= \beta_0 \left[ 1 + \sin\left(\frac{2\pi x}{L_x}\right) \right] , \\
\beta^2(x,y) &= \beta_0 \left[ 1 + \sin\left(\frac{2\pi x}{L_x}\right) \sin\left(\frac{2\pi y}{L_y}\right) \right] ,
\end{aligned}
\tag{20}
$$

where $\beta_0$ is a chosen non-dimensional constant. Differences may arise depending on the prescribed values for the parameters

$\alpha$, $L_x$, $L_y$ and $\beta_0$. Experiment 2 represents the ISMIP experiments C and D for $L = 10$ km (Pattyn et al., 2008), but in our case using non-dimensional variables.

The mechanical part of Experiment 3 is analogous to Experiment 2. The boundary conditions are periodic in $x$ and $y$ directions. The thermal problem requires additional boundary conditions in terms of temperature or fluxes. We set the surface temperature $T_0$ to 0. At the bottom, we set the vertical flux $q_z$ to 0 and, on the sides, we impose periodic boundary conditions.



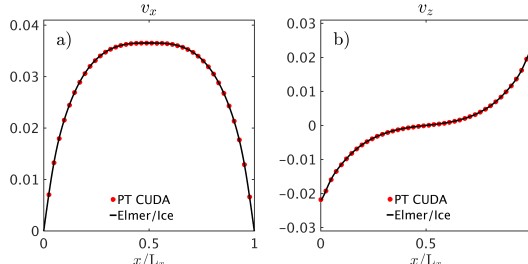

**Figure 4.** Comparison of the non-dimensional simulation results for the 2-D configuration of Experiment 1. We show a) the horizontal component of the surface velocity, $v_x$, and b) the vertical component of surface velocity, $v_z$, across the ice slab for both our PT GPU-based model and Elmer/Ice. For context, the maximum horizontal velocity ($v_x \approx 0.0365$) corresponds to $\approx 174$ m/yr. The horizontal distance is 2 km, while the ice thickness is 200 m. The box is inclined at $10°$.

The model parameters used in Experiment 3 are compiled in Table 2. We employ the semi-analytical 1-D model (Section 2.3) as an independent benchmark for the Experiment 3 calculations.

## 5 Results and performance

### 5.1 Experiment 1: Stokes flow without basal sliding

We compare our numerical solutions obtained with the GPU-based PT method using a CUDA C implementation to the reference Elmer/Ice model. We report all the values in their non-dimensional form, and the horizontal axes are scaled with their aspect ratio. In Figure 4, we plot both horizontal $v_x$ and vertical $v_z$ velocity components at the top surface for Experiment 1 in 2-D. Since the horizontal velocity component vanishes at the left and right boundary, $v_x = 0$, the maximum velocity values in the horizontal direction are located in the middle of the slab. We impose a no-slip boundary condition on all velocity components at the base and prescribe free-slip boundary conditions on all lateral domain sides. We prescribe a stress-free upper boundary in the vertical direction. On the left side ($x/\mathrm{L}_x = 0$), the ice is pushed down (compression); thus, the vertical velocity values were negative. On the right side ($x/\mathrm{L}_x = 1$), the ice is pulled up (extension), and the vertical velocity values were positive. Our PT GPU-based results agree well with the numerical solutions produced by Elmer/Ice. The numerical resolution of the Elmer/Ice model is $1001 \times 275$ grid-points in $x$ and $z$ directions ($\approx 8.25 \times 10^5$ degrees of freedom (DOF)), while we employed $2047 \times 511$ grid-points ($\approx 3.13 \times 10^6$ DOF) within our PT method. The PT method's efficiency enables considering the large number of grid-points without affecting the runtime. The DOF represent three variables in 2-D ($v_x, v_z, P$) and four variables in 3-D ($v_x, v_y, v_z, P$) multiplied by the number of grid-points involved.

Figure 5 shows the results for the 3-D configuration of Experiment 1. It plots our computed horizontal $v_x$, $v_y$ and vertical $v_z$ velocity components at the top surface (Figure 5a,c,e) and compares them to the reference solution from Elmer/Ice at $y \approx \mathrm{L}_y/4$ (Figure 5b,d,f). We find good agreement between the two model solutions. We employed a numerical resolution grid resolution



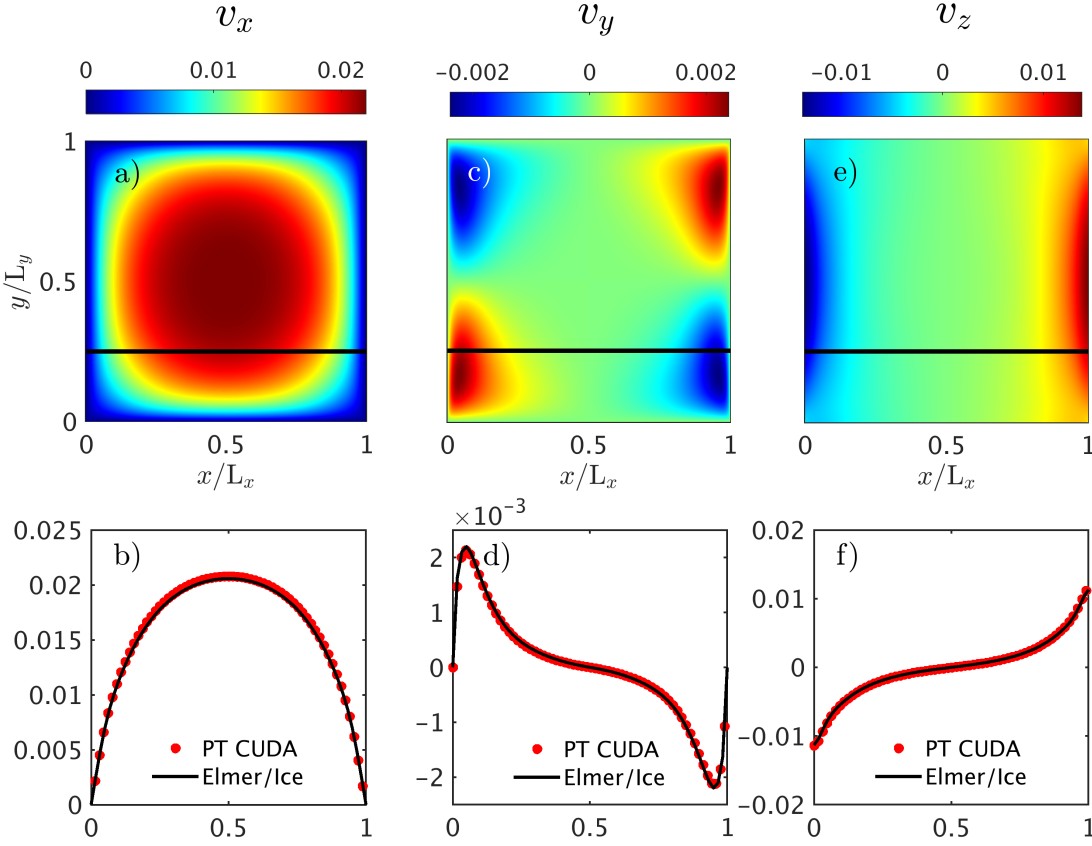

**Figure 5.** Non-dimensional simulation results for the 3-D configuration of Experiment 1. We report a) the horizontal surface velocity component $v_x$, c) the horizontal surface velocity component $v_y$, and e) the vertical surface velocity component $v_z$. The black solid line depicts the position where $y = \mathrm{L}_y/4$. Panels b) d) and f) show the surface velocity components $v_x, v_y$ and $v_z$, respectively, at $y = \mathrm{L}_y/4$ and compare them against the results from the Elmer/Ice model.

of $319 \times 159 \times 119$ grid-points in $x$, $y$ and $z$ directions ($\approx 2.41 \times 10^7$ DOF), and used a numerical grid resolution of $61 \times 61 \times 21$ ($\approx 3.1 \times 10^5$ DOF) in Elmer/Ice. Scaling our result to dimensional values (Table 1) results in maximal horizontal velocity ($v_x$) of $\approx 105$ m/yr. The horizontal distance is 2 km in the $x$-direction and 800 m in the $y$-direction, and the ice thickness is 200 m. The box is inclined of $10°$.

### 5.2    Experiment 2: Stokes flow with basal sliding

We now consider the case where ice is sliding at the base (ISMIP experiments C and D). We prescribe periodic boundary conditions at the lateral boundaries and apply a linear sliding law at the base. The top boundary remains stress-free in the

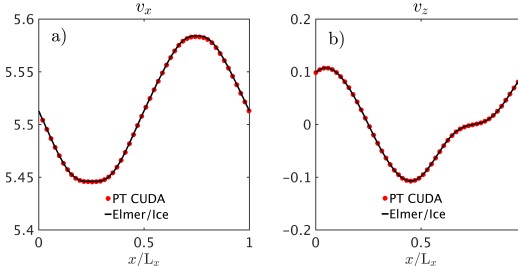

**Figure 6.** Non-dimensional simulation results for the 2-D configuration of Experiment 2. We plot a) the horizontal surface velocity component $v_x$ and b) the vertical surface velocity component $v_z$ across the slab for both our PT GPU-based model and Elmer/Ice. In dimensional terms, the maximum horizontal velocity ($v_x \approx 5.58$) corresponds to $\approx 16.9$ m/yr. The horizontal distance is 10 km, while the ice thickness is 1 km. The box is inclined at $0.1°$.

vertical direction. Figure 6 shows the results of the 2-D simulation of Experiment 2, where we employed a numerical grid resolution of $511 \times 127$ grid-points ($\approx 1.95 \times 10^5$ DOF) for the PT GPU-based solver and computed the Elmer/Ice solution using a numerical grid resolution of $241 \times 120$ ($\approx 8.7 \times 10^4$ DOF). We show both $v_x$ and $v_z$ velocity components at the slab's

surface. The two models' results agree well.

The 3-D simulation results for Experiment 2 appear in Figure 7. The upper panels (Figure 7a,c,e) show the spatial pattern in the three surface velocity components $v_x, v_y$ and $v_z$ computed with our PT GPU-based solver. The lower panels (Figure 7b,d,f) compare the three surface velocity components at $y \approx L_y/4$ computed by our PT GPU-based solver to Elmer/Ice. We employed a numerical grid resolution of $256 \times 256 \times 64$ ($\approx 1.67 \times 10^7$ DOF) for our PT GPU-based solver and a numerical

grid resolution of $61 \times 61 \times 21$ ($\approx 3.12 \times 10^5$ DOF) in the Elmer/Ice model. In dimensional units, the maximum horizontal velocity ($v_x$) corresponds to $\approx 16.4$ m/yr. The horizontal distance is 10 km in the $x$-direction 10 km in the $y$-direction, and the ice thickness is 1 km. The box is inclined at $0.1°$.

We find good agreement between the two numerical implementations, despite some discrepancies in the horizontal velocity component $v_y$. A potential explanation for the minor mismatch is the fact that the finite-element grid does not exactly coincide

with the location $y = L_y/4$ in Elmer/Ice, which may be resolved by specifically pinning nodes of the finite-element mesh. Since the flow is mainly oriented in the $x$ direction, the $v_y$ velocity component is more than two orders of magnitude smaller than the $v_x$ velocity component. Numerical errors in $v_y$ are more apparent than in the leading velocity component $v_x$. We report a one-order magnitude increase in the time-to-solution in Experiment 2 compared to the Experiment 1 configuration owing to the periodicity on the lateral boundaries.

**5.3   Experiment 3: Thermomechanically coupled Stokes flow without basal sliding**

We first verify that both the 1-D, 2-D and 3-D model configurations from Experiment 3 produce identical results assuming periodic boundary conditions on all lateral sides. In this case, all the variations in the $x$ or $y$ directions vanish ($\partial/\partial x$ and $\partial/\partial y$); thus, both the 2-D and 3-D models reduce to the 1-D problem. We employ a numerical grid resolution of $127 \times 127 \times 127$ grid-

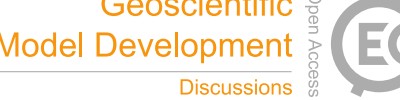



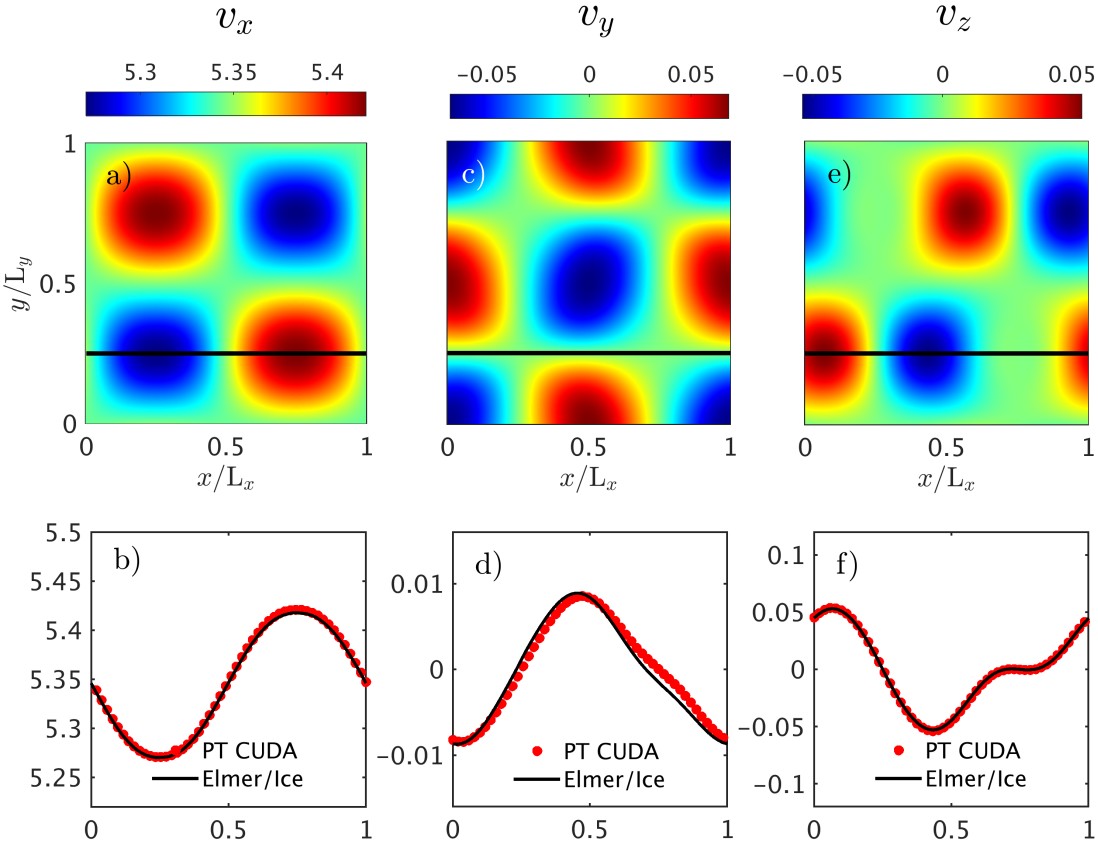

**Figure 7.** Non-dimensional simulation results for the 3-D configuration of Experiment 2. We report a) the horizontal surface velocity component $v_x$, c) the horizontal surface velocity component $v_y$ and e) the vertical surface velocity component $v_z$. The black solid line depicts the position where $y = L_y/4$. Panels b) d) and f) show the surface velocity components $v_x, v_y$ and $v_z$, respectively, at $y = L_y/4$ and compare them against the results from the Elmer/Ice model.

points in $x$, $y$ and $z$ direction, $127 \times 127$ grid-points in $x$ and $z$ directions and 127 grid-points in the $z$ direction for the 3-D,

2-D and 1-D problems, respectively.

We ensure that all results collapse onto the semi-analytical 1-D model solution (Section 2.3), which we obtained by analytically integrating the velocity field and solving the decoupled thermal problem separately (Eq. 11). From a computational perspective, we numerically solve Eq. 11 using a high spatial and temporal accuracy and therefore minimise the occurrence of numerical errors. We establish the 1-D reference solution for both the temperature and the velocity profile, solving Eq. 11

on a regular grid, reducing the physical time steps until we converge to a stable reference solution. Our reference simulation



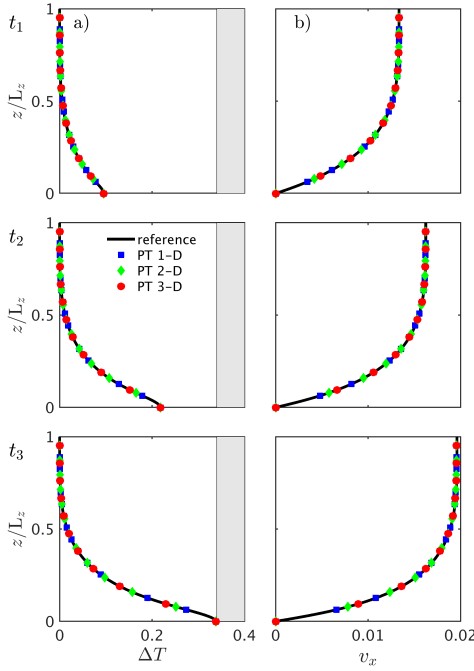

**Figure 8.** Non-dimensional simulation results for a) the temperature deviation $T$ and b) the horizontal velocity component $v_x$ for the 1-D, 2-D and 3-D PT GPU-based models at three different non-dimensional times $0.7 \times 10^8$, $1.4 \times 10^8$ and $1.9 \times 10^8$ and compare them to the 1-D reference model results. We employ a vertical grid resolution $n_z$ of $31, 95$ and $201$ grid-points. We sample the 1-D profiles at location $x = \mathrm{L}_x/2$ in 2-D and at $x = \mathrm{L}_x/2$ and $y = \mathrm{L}_y/2$ in 3-D. The shaded areas correspond to the part of the solution that is above the melting temperature, since we do not account for phase transitions in this case.

involves $4000$ grid-points and a non-dimensional time step of $5 \times 10^5$ (using a backward Euler time integration). We reach the total simulation time of $2.9 \times 10^8$ within $580$ physical time steps.

We report overall good agreement of all model solutions (1-D, 2-D, 3-D and 1-D reference) at the three reported stages for this scenario (Figure 8). As expected from the 1-D model solution, temperature varies only as a function of time and depth with
the highest value obtained close to the base and for longer simulation times. Similarly, the velocity profile is equivalent to the 1-D profile and the largest velocity value is located at the surface. We only report the horizontal velocity component $v_x$ for the 2-D and the 3-D models, since $v_y$ and $v_z$ feature negligible magnitudes. Thus, we only observe spatial variation in the vertical $z$ direction. We report the non-dimensional temperature $T$ (Figure 9a) and horizontal velocity $v_x$ (Figure 9b) fields for both the 3-D and the 2-D configurations compared at non-dimensional time $0.7 \times 10^8$, $1.4 \times 10^8$ and $1.9 \times 10^8$. The dimensional results
from Experiment 3 correspond to a 300 m thick ice slab inclined at $10°$ angle with an initial surface temperature of -10°C. The maximum initial velocity for the isothermal ice slab corresponds to $\approx 486 \, \mathrm{m/yr}$, while the maximum velocity just before the melting point is reached corresponds to $830 \, \mathrm{m/yr}$. The comparison snapshot times are $1.6$, $3.2$ and $4.4$ years.

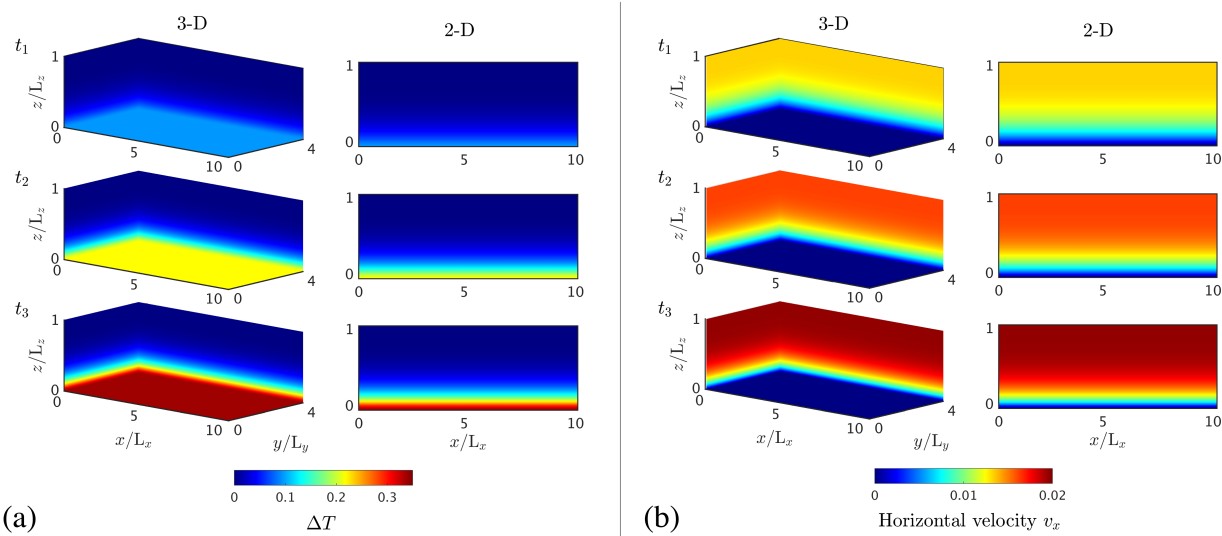

**Figure 9.** Spatial distribution of a) the temperature deviation from the initial temperature $T$ and b) the horizontal velocity component $v_x$ for the 3-D (left column) and the 2-D (right column) in non-dimensional units. We scale the domain extend with $L_z$. We compare the numerical solutions at non-dimensional times $0.7 \times 10^8$, $1.4 \times 10^8$ and $1.9 \times 10^8$.

The semi-analytical 1-D solution enables us to evaluate the influence of the numerical coupling method and time integration and to quantify when and why high spatial resolution is required in thermomechanical ice flow simulations. We compare the

1-D semi-analytical reference solution (Eq. 11) to the results obtained with the 1-D PT-based solver for three spatial numerical resolutions ($n_z =$31, 95 and 201 grid-points) at three non-dimensional times $1 \times 10^8$, $2 \times 10^8$ and $2.9 \times 10^8$ (Figure 10). The grey area in Figure 10 highlights where the melting temperature is exceeded. Since our semi-analytical reference solution does not include phase transitions, we also neglect this component in the numerical results. During the early stages of the simulation, the thermomechanical coupling is still minor and solutions at all resolution levels are in good agreement with one another and

with the reference. The low resolution solution starts to deviate from the reference (Figure 10b) when the coupling become more pronounced close to the thermal runaway point (Clarke et al., 1977). The high spatial resolution solution is satisfactory at all stages. We conclude that high spatial resolutions is required to accurately capture the non-linear coupled behaviour in regimes close to the thermal runaway, which is seldom the case in the models reported in the literature.

     Thermomechanical strain localisation may significantly impact on the long-term evolution of a coupled system. A recent

study by Duretz et al. (2019) suggested that partial coupling may result in under-estimating the thermomechanical localisation compared to the fully coupled approach, as reported in their Figure 8. We compare three coupling methods (Figure 11): (1) A fully coupled implicit PT method, as described in the numerical section, where the viscosity and the shear-heating term are implicitly determined by using the current guess. (2) An implicit numerically uncoupled mechanical and thermal model. (3) An explicit numerically uncoupled mechanical and thermal model. The numerical time integration in physical time is performed

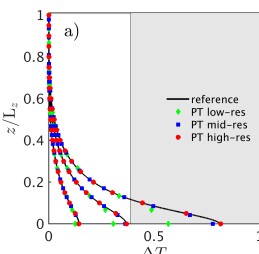
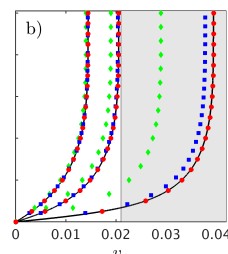

**Figure 10.** Non-dimensional simulation results for a) the temperature deviation $T$ and b) the horizontal velocity component $v_x$ to test solver performance at three resolutions. The vertical resolutions are $\mathrm{LR} = 31$, $\mathrm{MR} = 95$ and $\mathrm{HR} = 201$ grid-points for low-, mid- and high-resolution runs, respectively. We compare the results for non-dimensional time $1 \times 10^8$, $2 \times 10^8$ and $2.9 \times 10^8$. The shaded areas correspond to the part of the solution that is above the melting temperature, since we do not account for phase transitions in this benchmark.

using an implicit backward Euler method for (1) and (2) and a forward Euler explicit time integration method for (3). We utilise the identical non-dimensional time step for both the explicit and the implicit numerical time integration. We perform 580 time steps, reaching a simulation time of $2.9 \times 10^8$. We employ a vertical grid resolution of $n_z = 201$ grid-points for all models. The chosen time step for the explicit integration of the heat diffusion equation is below the CFL stability condition given by $\Delta z^2/2.1$ in 1-D, where $\Delta z$ represent the grid spacing in a vertical direction.

Physically, the viscosity and shear-heating terms are coupled and are a function of temperature and strain-rates, but we update the viscosity and the shear-heating term based on temperature values from the previous physical time step. Thus, the shear-heating term can be considered as a constant source term in the temperature evolution equation during the time step, leading to a semi-explicit rheology. We show the 1-D numerical solutions of (blue) the fully coupled method with a backward Euler (implicit) time integration and the two uncoupled methods with either (green) backward (implicit) or (red) forward (explicit)

Euler time integration (Figure 11) and compare them to the 1-D reference model solution. Surprisingly, and in contrast to Duretz et al. (2019), we observe a good agreement between all methods, suggesting that the different coupling strategies capture the coupled flow physics with sufficient accuracy given high enough spatial and temporal resolution. However, for a longer-term evolution, the uncoupled approaches may predict lower temperature and velocity values than the fully coupled approach.

**5.4   Experiment 3: Thermomechanically coupled Stokes flow in a finite domain**

Boundary conditions corresponding to immobile regions in the computational domain may induce localisation of deformation and flow observed in locations such as shear margins, grounding zones or bedrock interactions. Dimensionality plays a key role in such configurations, causing the stress distribution to be variable among the considered directions.

We used the configuration in Experiment 3 to investigate the spatial variations in temperature and velocity distributions by defining no-slip conditions on the lateral boundaries for the mechanical problem and hindering any heat flux through those

boundaries. We employ a numerical grid resolution of $511 \times 255 \times 127$ grid-points, $511 \times 127$ grid-points and 201 grid-points





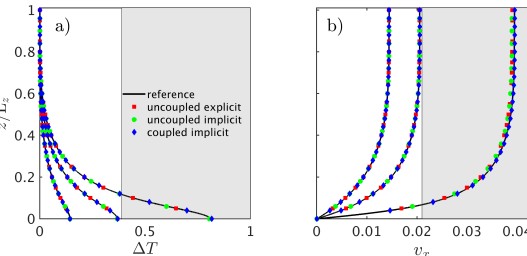

**Figure 11.** Non-dimensional simulation results for a) the temperature deviation $T$ and b) the horizontal velocity component $v_x$ to evaluate different numerical time integration schemes. We consider three non-dimensional time $1 \times 10^8$, $2 \times 10^8$ and $2.9 \times 10^8$ and compare our numerical estimates to the reference model. As before, the shaded areas correspond to the part of the solution that is above the melting temperature, since we neglect phase transitions in this comparison.

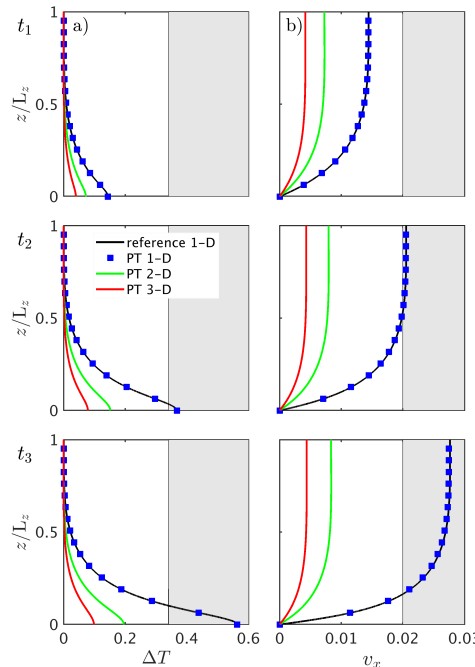

**Figure 12.** Non-dimensional simulation results for a) the temperature deviation $T$ and b) the horizontal velocity component $v_x$ for the 1-D, 2-D and 3-D PT GPU-based models at three non-dimensional times $1 \times 10^8$, $2 \times 10^8$ and $2.5 \times 10^8$ compared to our analytical solution. We sample the 1-D profiles at location $x = L_x/2$ in 2-D and at $x = L_x/2$ and $y = L_y/2$ in 3-D. The shaded area corresponds to the part of the solution that is above the melting temperature, approximately 0.35 of the temperature deviation.

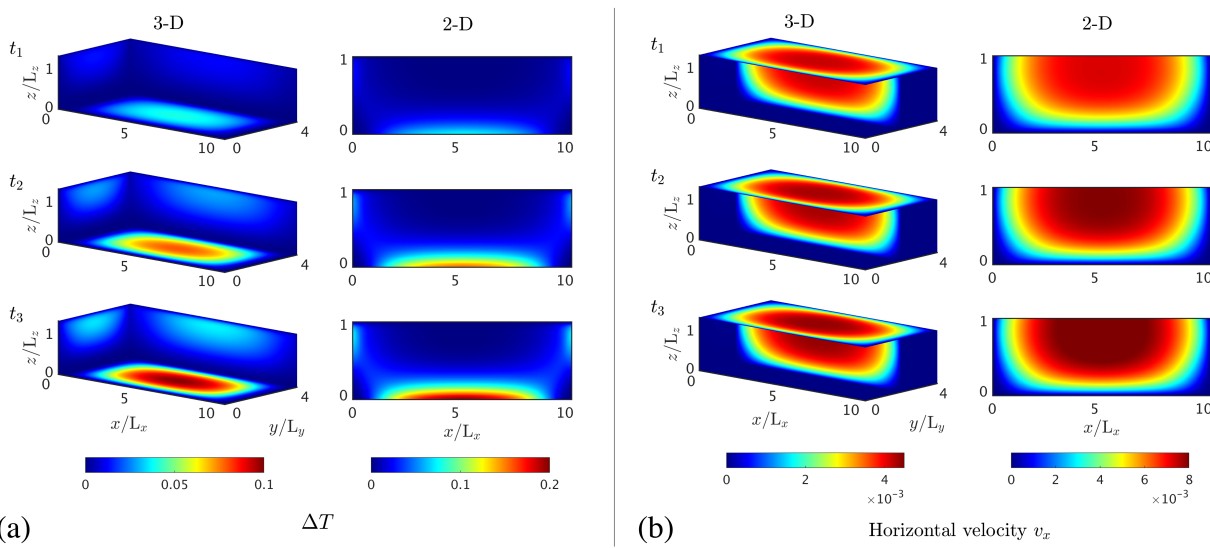

**Figure 13.** Non-dimensional simulation results of a) the temperature deviation from the initial temperature $T$ and b) the horizontal velocity component $v_x$ for Experiment 3 at three non-dimensional times $1 \times 10^8$, $2 \times 10^8$ and $2.5 \times 10^8$ for both the 2-D and 3-D configurations.

for the 3-D, 2-D and 1-D case, respectively. We prescribe a non-dimensional time step of $5 \times 10^5$. We perform 500 numerical time steps and reach a total non-dimensional simulation time of $2.5 \times 10^8$. We then compare the temperature $T$ and horizontal velocity component $v_x$ at three times obtained with the 1-D, 2-D and 3-D PT GPU-based solver to the reference solution (Figure 12). We use 1-D profiles for comparison, taken at location $x = L_x/2$ in the 2-D model and at location $x = L_x/2$ and

$y = L_y/2$ in the 3-D model. We also report the temperature variation $\Delta T$ (Figure 13a) and the horizontal velocity component $v_x$ (Figure 13b) for both the 2-D and 3-D simulations. The melting temperature approximately corresponds to 0.35 of the temperature deviation. The reported results correspond to a $2.3-$, $4.6-$ and $5.8-$ year evolution.

     All three models start with identical initial conditions for the thermal problem, i.e. $\Delta T = 0$ throughout the entire ice slab. The difference between the models arises owing to different stress distributions in 1-D, 2-D or 3-D. For instance, the additional

stress components inherent in 2-D and 3-D are in the same order of magnitude as the 1-D shear stress for the considered aspect ratio, reducing the horizontal velocity $v_x$ in the 2-D and 3-D models. This also impacts on the shear-heating term, reducing the source term in the temperature evolution equation. In the 1-D configuration, the unique shear stress tensor component is a function only of depth. On the other end-member, the 3-D configurations allow for a spatially more distributed stress state. They lower strain-rates in this scenario and reduce the magnitude of shear-heating in higher dimensions. The spatially heterogeneous

temperature and strain-rate fields in all directions require the utilisation of sufficiently high spatial numerical resolution in all directions in order to accurately resolve spontaneous localisation.

     We did not consider phase transition in the previous experiments for the sake of model comparison and because the analytical solution excluded this process. The existence of a phase transition caps the temperature at the pressure melting point in regions

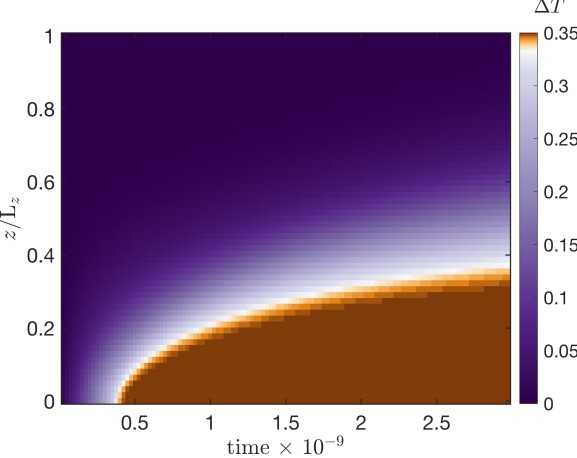

**Figure 14.** Experiment 3 includes a phase transition owing to melting. We report the evolution in time of non-dimensional temperature variation $\Delta T$ along a vertical profile picked at location $x = L_x/2$ within a 2-D run from Experiment 3. For this purpose, we run the 2-D PT GPU-based models from Experiment 3 for a duration of $2.9 \times 10^9$.

with pronounced shear-heating, as illustrated in 2-D in Figure 14. The simulation represents the thermomechanically coupled
Experiment 3 with no-sliding and heat impermeable walls (similar to Figure 13). Meltwater production consumes excess heat generated by shear-heating. Thus, melting provides a physical mechanism that avoids thermal runaway in shear-heating dominated zones in the ice. The experiment duration in dimensional units is 70 years, and the maximal temperature increase is $10°C$ upon reaching the melting point.

## 5.5    The computational performance

We used two metrics to assess the performance of the developed PT algorithm: the effective memory throughput ($\mathrm{MTP_{eff}}$) and the wall-time. We first compare the effective memory throughput of the vectorised MATLAB CPU implementation and the single-GPU CUDA C implementation. We employ double-precision (DP) floating-point arithmetic in CUDA C for fair comparison. Second, we employ the wall-time metric to compare the performance of our various implementations (MATLAB, CUDA C) and compare these to the time-to-solution of the Elmer/Ice solver.

We use two methods to solve the linear system in Elmer/Ice. In the 2-D experiments, we use a direct method and in 3-D, an iterative method. The direct method used in 2-D relies on the UMFPACK routines to solve the linear system. To solve the 3-D experiments, we employ the available bi-conjugate gradient stabilised method (BICGstab) with an ILU0 preconditioning. We employ the configuration in Experiment 1 for all the performance measurements. We use an Intel i7 4960HQ 2.6 GHz (Haswell) four-core CPU to benchmark all the CPU-based calculations. For simplicity, we only ran single-core CPU tests,
staying away from any CPU parallelisation of the algorithms. Thus, our MATLAB or the Elmer/Ice single-core CPU results are not representative of the CPU hardware capabilities, and are only reported for reference.

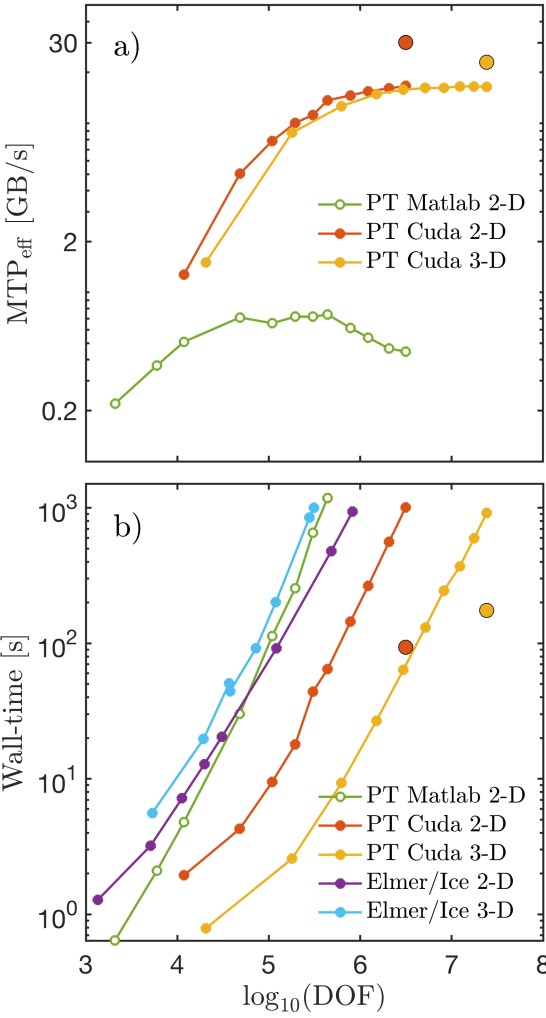

**Figure 15.** Performance evaluation of the PT mechanical solver in terms of: a) the effective memory throughput $\mathrm{MTP_{eff}}$ in GB/s and b) the wall-time (in seconds) to converge the Stokes solver to a relative non-linear tolerance of $\mathrm{tol_{nonlin}} = 10^{-8}$. We report the results obtained using a 2-D PT CPU-based single-core vectorised MATLAB implementation, a 2-D and 3-D PT GPU-based CUDA C implementation and a 2-D (direct) and 3-D (iterative) solver within the Elmer/Ice FEM single-core CPU-based model. The CPU codes are executed on an Intel i7 4960HQ CPU processor with 8 GB RAM, and the GPU codes are launched on an Nvidia Titan X (Maxwell) GPU with 12 GB on-board memory. All the computations are performed in double-precision arithmetic, with the only exception for the two single-precision GPU-based runs depicted with larger red (2-D) and orange (3-D) symbols. The single-core CPU PT MATLAB and Elmer/Ice results are shown for reference; they are not meant for performance comparison because we did not enable multi-threading in MATLAB and did not have access to a parallel version of Elmer/Ice.





The PT solver relies on evaluating a finite-difference stencil. Each cell of the computational domain needs to access neighbouring values in order to approximate derivatives. These memory access operations are the performance bottleneck of the algorithm, making it memory-bounded. Thus, the algorithm's performance depends crucially on the memory transfer speed, and not the rate of the floating-point operations. Memory-bounded algorithms place additional pressure on modern many-core processors, since the current chip design tends to large flop-to-byte ratios. Over the past years and decades, the memory bandwidth increase has been much slower compared to the increase in the rate of floating-point operations.

As shown by Omlin (2017) and Räss et al. (2019a), a relevant metric to assess the performance of memory-bounded algorithms is the effective memory throughput ($\mathrm{MTP_{eff}}$) (Eq. 21). The $\mathrm{MTP_{eff}}$ determines how efficiently data is transferred between the main memory and the arithmetic units and is inversely proportional to the execution time:

$$\mathrm{MTP_{eff}} = \frac{(n_x n_y n_z) n_{\mathrm{iter}}\, n_{\mathrm{IO}}\, n_{\mathrm{p}}}{1024^3\, t_{\mathrm{nt}}} \qquad [\mathrm{GB/s}] \tag{21}$$

where $(n_x n_y n_z)$ stands for the total number of grid-points, $n_{\mathrm{iter}}$ is the total number of numerical iterations performed, $n_{\mathrm{p}}$ is the arithmetic precision (single – 4 bytes or double – 8 bytes), $t_{\mathrm{nt}}$ is the wall-time in seconds needed to compute the $n_{\mathrm{iter}}$ iterations, and $n_{\mathrm{IO}}$ is the performed number of memory accesses. It represents the minimum number of memory operations (read-and-write or read only) required to solve a given physical problem. For instance, in the mechanical Stokes solver for Experiment 1, we have to update (read-and-write) three arrays ($v_x, v_z$ and $P$) at every iteration in 2-D and four arrays ($v_x, v_y, v_z$ and $P$) at every iteration in 3-D. Thus, the update of the mandatory arrays requires a minimum of six (eight) read-and-write operations in 2-D (3-D). One additional read-and-write is needed to resolve the non-linear viscosity; thus, $n_{\mathrm{IO}} = 10$ in 2-D case and $n_{\mathrm{IO}} = 12$ in 3-D.

We report $\mathrm{MTP_{eff}}$ values obtained with the PT algorithm for both the vectorised MATLAB (CPU) and the CUDA C (GPU) implementations in double-precision arithmetic (Figure 15a). We also show the GPU performance using single-precision arithmetic (Figure 15a – green diamonds). The results we obtain should be compared to the peak memory throughput value $\mathrm{MTP_{peak}}$ for the specific hardware used. The $\mathrm{MTP_{peak}}$ reports the memory transfer rates delivered only by performing memory copy operations with no computations. This value reflects the hardware performance limit and the maximal effective memory bandwidth. We measure $\mathrm{MTP_{peak}}$ values for the Intel i7 4960HQ CPU of 20 GB/s, and of 260 GB/s for the Nvidia Titan X GPU. The single-core vectorised MATLAB CPU implementation achieves about 0.7 GB/s, and the CUDA C implementation 16 GB/s. Thus, the MATLAB single-core CPU implementation reaches 3.5% of the (CPU) hardware peak value, and the CUDA C (GPU) implementation at about 6.15% and 11% of the (GPU) hardware peak value using double-precision and single-precision arithmetic, respectively. Further improvement of the GPU $\mathrm{MTP_{eff}}$ values can be achieved by optimising the GPU code using more on-the-fly calculations and advanced kernel scheduling.

We investigate the wall-time to solve one time step with the PT GPU-based solver for both the 2-D and the 3-D configurations (Figure 15b). We found wall-times of about 15 minutes to solve $\approx 2.4 \times 10^7$ DOFs with double-precision arithmetic and only three minutes when using single-precision arithmetic on a Nvidia Titan X (Maxwell) GPU. In future investigations, one may consider comparing wall-times obtained by CPU algorithms fully enabling all cores of the CPU against wall-times for GPUs within the same price and power consumption range.



The 3-D performance results obtained on various available Nvidia GPUs are summarised in Figure 16). We performed all the calculations using double-precision arithmetic. We compare the $\text{MTP}_\text{eff}$ and wall-time values as functions of the DOF. We tested GPUs from various price ranges and chip generations, targeting entry-level GPUs such as the Nvidia Quadro P1000 (Pascal), high-end gaming cards such as the Nvidia Titan Black (Kepler) or the Nvidia Titan X (Maxwell), and data-centre-

class GPU accelerators such as the Nvidia Tesla V100 PCIe (Volta). The MATLAB implementation peak $\text{MTP}_\text{eff}$ values are about $0.46$ GB/s, the Quadro P1000 (Pascal) values about $4.3$ GB/s, the Titan Black (Kepler) $12.4$ GB/s, the Titan X (Maxwell) $16.7$ GB/s, and the Tesla V100 (Volta) $83.2$ GB/s. The $\text{MTP}_\text{eff}$ values directly impact on the wall-time, since the memory bandwidth was the bottleneck. We solved a 3-D problem involving $511 \times 255 \times 127$ grid-points ($6,6 \times 10^7$ DOF) in about one hour on the Titan Black GPU, 40 minutes on the Titan X GPU, and only eight minutes on the Tesla V100 GPU. Notably, at

this resolution, we employed about 4.5 GB of memory to solve the isothermal Stokes model. The results suggest that more recent GPUs such as the data-centre Tesla V100 (Volta) offer a significant (order of magnitude higher) performance increase than entry-level GPU accelerators, such as the Quadro P1000.

We share the performance of the GPU-MPI implementation of our solver to execute on distributed memory machines. We achieved a weak scaling parallel efficiency of 93% on the 128 Nvidia Titan X (Maxwell) GPUs on the *octopus* supercomputer

at the Swiss Geocomputing Centre, University of Lausanne, Switzerland. As baseline, we employed a non-MPI single GPU calculation. We then repeated the experiment using 1 to 128 MPI (thus GPUs) processes and report the normalised execution time (Figure 17). The effective drop in parallel efficiency is only 4% involving 1 to 128 MPI processes. We achieved this close-to-optimal parallel efficiency by overlapping MPI message communication and local domain stencil calculations. We specifically employed a CUDA stream in order to execute the communication and computation overlap asynchronously. We

performed similar experiment on the *volta* node, an 8 Nvidia Tesla V100 32 GB (Nvlink Volta) based computer node (analogous to Nvidia's DGX-1 box), reporting a parallel efficiency of 0.985% for a single MPI process running at 0.99% of the non-MPI reference.

## 6   Discussion

Numerically resolving thermomechanical processes in ice is vital for improving our understanding of the complex behaviour of

ice sheets and glaciers. To date, very few studies have investigated the numerical aspects of thermomechanically coupled Stokes solvers (e.g., Duretz et al., 2019). Existing assessments (e.g., Zhang et al., 2015) usually employed low spatial resolution, and did not address the influence of the numerical implementation of multi-physics coupling strategies or the role of numerical time integration. To avoid the significant computational expense of a thermomechanically coupled full Stokes model, many studies relied either on the computationally less expensive shallow ice approximations, linear or linearised Stokes models,

or low spatial resolutions. None of the approaches have resolved the multi-physics and multi-scale processes governing the boundaries of streaming ice, including shear margins, grounding zones and the basal interface.

To address these limitations, we have developed a new numerical model based on an iterative pseudo-transient finite-difference method. Our discretisation yields to a concise matrix-free algorithm well suited to use the intrinsic parallelism

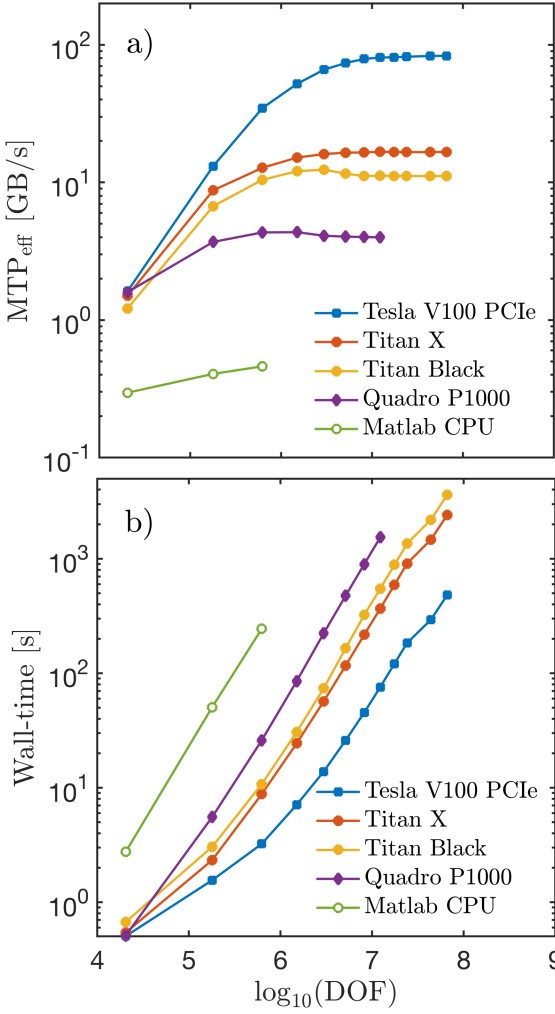

**Figure 16.** Performance evaluation of the PT mechanical solver in terms of: a) effective memory throughput $\mathrm{MTP_{eff}}$ in GB/s and b) wall-time (in seconds) to converge the Stokes solver to a relative non-linear tolerance of $\mathrm{tol_{nonlin}} = 10^{-8}$. We report the results obtained using a 3-D PT CPU-based single-core vectorised MATLAB implementation and a 3-D PT GPU-based CUDA C implementation running on different GPU chip architectures. The CPU codes are executed on an Intel i7 4960HQ CPU processor with 8 GB RAM. The GPU codes were launched on an Nvidia Titan Black (Kepler) GPU with 6 GB, an Nvidia Titan X (Maxwell) GPU 12 GB, an Nvidia Quadro P1000 (Pascal) 4 GB and an Nvidia Tesla V100 PCIe (Volta) 32 GB.

of modern hardware accelerators such as GPUs. Our choices enable high-resolution 2-D and 3-D thermomechanically cou-
pled simulations to efficiently perform on desktop computers and to scale linearly on supercomputers, both featuring GPU
accelerators.

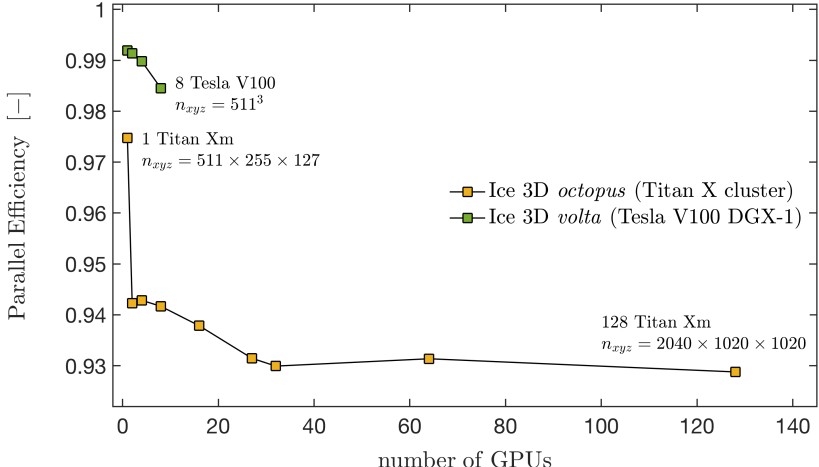

**Figure 17.** MPI weak scaling of the 3-D thermomechnically coupled GPU-based ice software. We report the parallel efficiency [-] of the numerical application on two Nvidia hardware accelerators, the 1-128 Titan X (Maxwell) 12 GB consumer electronics GPUs and the 1-8 Tesla V100 32 GB Nvlink data-centre GPUs. Both are available via the *octopus* supercomputer and the *volta* node, respectively. Note that the execution time baseline used to compute the parallel efficiency represents a non-MPI calculation.

The significant temperature dependence of ice's shear viscosity leads to pronounced spatial variations in the viscosity, which affects the convergence rate of our iterative PT method. Resolving shear flow localisation is challenging in this context, since it requires the simultaneous minimisation of errors in locations of the computational domain that are governed by different characteristic time scales. Our PT approach allows us to capture the resulting spatial heterogeneity and offers a physically-motivated strategy to locally ensure stability of the iterative scheme using local pseudo-time steps, analogous to diagonal preconditioning in matrix-based direct approaches. The conciseness and simplicity of the implementation allows us to explore influences of various coupling methods and time integrations in a straight-forward way. Similar arguments suggest that the PT approach is an interesting choice for educational purposes.

We quantify the scalability of our approach through extensive performance tests, where we investigated both the time-to-solution and the efficiency of exploiting the current hardware capabilities at their maximal capacities. To verify the accuracy and the coherence of the proposed results, we performed a set of benchmark experiments, obtaining excellent agreement with results from the widely used glacier flow model Elmer/Ice. Experiment 3 verifies that, under the assumption of periodic configurations, both 1-D, 2-D and 3-D models return matching results.

Further, we have tested the accuracy of our numerical solutions for different time integration schemes, including forward (explicit) and backward (implicit) Euler and different physical time steps. The value of the numerical time step must be chosen as sufficiently small so as to resolve the relevant physical processes. We limited the maximal time step in the explicit time integration scheme by the CFL stability criterion for temperature diffusion. For high spatial numerical resolutions, the CFL-based time step restriction is sufficient to resolve the coupled thermomechanical process. However, this conclusion is not valid



for low spatial resolutions (e.g., fewer than 20 grid-points). At low resolution, the CFL-based stability condition predicts time step values larger than the non-dimensional time ($2 \times 10^8$) needed to raise the temperature. Thus, we did not sufficiently resolve the physical process. An implicit scheme for the time integration remedies the stability issue, but does not guarantee accuracy. Independent of the numerical time integration scheme used, the range of time step values that resolve the coupled physics is close to the explicit stability criterion.

Our multi-GPU implementation of the thermomechanical PT solver achieved a close-to-ideal parallel efficiency featuring a runtime drop of only 4% compared to a single MPI process execution (a 7% deviation from a single non-MPI GPU runtime). We achieve this optimal domain decomposition parallelisation by overlapping communication and computation using native CUDA streams. This CUDA feature enables asynchronous compute kernel execution. Similar implementation and parallel scaling results were recently achieved for hydro-mechanical couplings (Räss et al., 2019a, b).

## 535 7 Conclusions

We have developed an iterative solver to efficiently exploit the capabilities of modern hardware accelerators such as GPUs. We report rapid execution times on single-GPUs monitoring and optimising memory transfers. We achieved a close-to-ideal parallel efficiency (93%) on a weak scaling test up to 128 GPUs by overlapping MPI communication and computations. We implemented the coupled thermomechanical PDEs using our iterative PT approach in a straight-forward way from the math-

ematical model. The technical advances and utilisation of GPU accelerators enabled us to investigate the thermomechanical coupling and to resolve the first-order physics governing the ice flow in 3-D on a high spatial and temporal resolution.

We benchmarked the mechanical solver of the coupled model against a community standard model Elmer/Ice in a set of experiments specifically designed to test the mechanical solver. We further investigated explicit and implicit coupling and time integration strategies. We report that the physical time step must be chosen with care. Sufficiently high temporal resolution is

mandatory in order to accurately resolve the coupled physics. Although minor differences arise among uncoupled and coupled approaches, we observe less localisation for uncoupled models compared to the fully coupled ones.

We established that a relatively high spatial numerical resolution is necessary to resolve the non-linear and spontaneous localisation of thermomechanically coupled ice flow, including more than 100 grid-points in the vertical direction. We stress that spatial variations in the horizontal plane can significantly impact on the ice flow dynamic, justifying high spatial numerical

resolution in all directions. We finally reported that considering the full 3-D stress tensor can significantly slow down the process of thermal runaway, which can ultimately be hindered by considering phase transitions.

GPUs are compact, affordable and relatively programmable devices that offer high performance throughput (close to TB/s peak memory throughput) and a good price to performance ratio. GPUs offer an attractive alternative to conventional CPUs owing to their massively parallel architecture featuring thousands of cores. The presented models lever this modern technology

and enable us to gain further process-based understanding of ice-flow localisation. Resolving the coupled processes at very high spatial and temporal resolutions provides future avenues to address current challenges in accurately predicting ice sheet dynamics.



*Code availability.* The FastICE software we used in this study is licensed under GPLv3 free software license. The latest version of the code is available for download from Bitbucket at https://bitbucket.org/lraess/ice/ and from http://wp.unil.ch/geocomputing/software/ice. Past

and future FastICE versions are available from a permanent DOI repository (Zenodo) at http://doi.org/10.5281/zenodo.3387669. The FastICE software includes code examples based on the PT method in both the MATLAB and CUDA C programming languages. The GPU routines require a CUDA-capable GPU device. The multi-GPU 3-D code require CUDA-aware MPI to be installed. On the ocotpus GPU supercomputer, we have CUDA 8.0 installed and built Open MPI 2.0.0 with CUDA 8.0, GCC 5.4 on a CentOS 6.9 system.

*Author contributions.* LR participated in the early model and numerical method development stages, implemented the MPI version of the

code, performed the scaling analysis, and reshaped the final version of the manuscript. AL realised the first version of the study, performed the benchmarks, and drafted the manuscript outline as the second chapter of his PhD thesis. FH and YP supervised the early stages of the study. JS contributed to the capped thermal model and provided feedback on the manuscript in the final stage. All authors have reviewed and approved off the final version of the manuscript.

*Competing interests.* The authors declare that they have no conflicts of interest.

*Acknowledgements.* We thank Dr. Samuel Omlin, Dr. Thibault Duretz and Mathieu Gravey for their technical and scientific support. We thank Dr. Thomas Zwinger for his valuable comments, which enhanced the study. We acknowledge the Swiss Geocomputing Centre for computing resources on the octopus supercomputer and are grateful to Philippe Logean for continuous technical support. LR acknowledges support from the Swiss National Science Foundation's Early Postdoc Mobility Fellowship 178075. This research was supported by the National Science Foundation through the Office of Polar Programs awards PLR-1744758 and PLR-1739027. This material is based upon

work supported by, or in part by, the U. S. Army Research Laboratory and the U. S. Army Research Office under contract/grant number W911NF-12-R0012-04.





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
