# Peer review of "Modelling thermomechanical ice deformation using a GPU-based implicit pseudo-transient method (FastICE v1.0)"

_Geoscientific Model Development, 2019_

## Referee Comment (RC1) · Anonymous Referee #1 · 12 Oct 2019

The authors present a new GPU-accelerated, 3D ice-sheet model and they test its performance against established benchmark tests from the ice-sheet modelling community. The new ice-sheet model is a full-Stokes variant, and will, in principle, be capable of handling the complex dynamics of rapidly flowing ice-streams and outlet glaciers.

The manuscript is well-written, and the governing equations are clearly presented. Overall the manuscript was enjoyable to read, and I learned a lot. The tests were also convincing – as convincing as visual comparisons of results can be. In general, the structure of the manuscript is traditional: Introduce new model, explain the basic principles and the implementation, and test the model against other models. This is

fine, and it provides a convenient reference for later work. However, as a reader I would have liked to see a demonstration of what the new model can really do – just a sneak peek into the suite of problems that the authors hope to address with this new model. There are so many ice models being presented, but it is unfortunately surprisingly rare that we see ice-sheet models applied in ways that make us wiser. So, if possible, I encourage the authors to include a demonstration of the model toward the end of the manuscript – something that is visually, and intellectually, more appealing than the benchmark tests.

Specific comments:

Line 36: The GPU-acceleration is very interesting and, as far as I know, rather new in ice-sheet models. However, a quick search leads to Braedstrup et al. (2014) "Ice-sheet modelling accelerated by graphic cards" in Computers & Geosciences 72, 210-220. This paper is not cited here, although it covers some of the same challenges and principles of GPU-acceleration.

line 42: Also, regarding GPU-acceleration, it would be good to see reference to other flow problems that have successfully been GPU accelerated. What problems and models have inspired the authors?

line 122: The comment on single-precision calculations leaves me confused. Are the GPU-calculations single precision? Or does it depend on the specific GPU architecture? Please clarify.

line 163: Braedstrup et al has a nice description of staggered grids and GPU acceleration – must be cited here.

line 175: Even up-wind advection schemes are going to suffer from numerical diffusion – and high numerical resolution is just making it worse. Please discuss this here.

line 182: The matrix-free solver using pseudo-time is nicely explained. However, it would be good to see exactly how the residuals propagate in the grid. Many similar

matrix-free relaxation schemes use multi-grid setups to make the residuals decay faster – these could be discussed.

Eqn. 15: I believe that theta < 1 is often referred to as underrelaxation.

Eqn. 19: Again, I miss information on how the residuals decay in the grid – particularly when using this stabilizing scheme. Also, I could not find previous reference to alpha, but I might have missed it.

line 299: I can see how the non-dimensional equation makes the implementation simpler, but is it necessary to present results in the non-dimensional form? It just makes the output harder to understand.

Section 5: There is some repetition of captions in the text. "In Figure 4, we plot..."; "Figure 5 shows..."; "Figure 6 shows..." etc. This could be skipped to make the text smoother.

line 308: Why are the benchmark tests performed at different resolutions? Does the GPU-model require order-of-magnitude more DOFs to yield the same accuracy as the FEM model? The comparisons give leave me with that impression, and then what is the advantage of the PT setup?

line 314: "numerical resolution grid resolution"

line 333: The authors are right to address the discrepancies between the model results – but why not follow up on the idea to pin nodes in the FEM mesh?

Fig. 15: The performance diagrams are very convincing – however, the use of widely different DOFs for the FEM and PT models in the benchmark tests makes we wonder if the speedup is real?

[Figure]

---

## Referee Comment (RC2) · Anonymous Referee #2 · 24 Oct 2019

This paper presents a new GPU-based thermomechanical land-ice dynamical core, termed FastICE. This dynamical core relies on a numerical framework in which pseudo-transient iterations solve the implicit thermomechanical coupling equations between the ice velocities (governed by the full Stokes equations with nonlinear viscosity given by Glen's flow law) and ice temperature. The spatial discretization for the governing equations is the finite difference method on a staggered Cartesian grid. The algorithm requires no preconditioned linear solves and no global reductions. Strong as well as weak scalability, including a 93% weak scaling parallel efficiency, is demonstrated for the GPU-accelerated code.

[Figure]

There is currently a lot of ongoing R&D in the area of performance portability of land ice (and more broadly climate) models to GPUs and other advanced architectures, but I have not seen any papers prior to this one that present and demonstrate a full end-to-end land-ice model that runs correctly and efficiently on GPUs. Hence, this paper has a lot of archival value, and I anticipate it will be very much of interest to glaciologists and computer scientists interested in performance portability of land ice models to GPUs. The proposed thermomechanical pseudotransient time-stepping formulation is nice in that it does not require a linear solve (the development of portable preconditioners/linear solvers for land-ice is an ongoing research area that is holding up certain land ice models from being fully ported). Additionally, the non-dimensionalization of the thermo-mechanically coupled full Stokes equations is a worthwhile contribution of the paper, even as something that stands alone from the GPU implementation (although the authors derive this primarily to allow them to study the effect of reduced precision arithmetic in their land ice computations). It has been argued by some researchers that running land-ice models non-dimensionally may reduce ill-conditioning and improve performance, as it often does in CFD. It is nice to have a non-dimensionalization documented in an archival publication such as this one.

Overall, this is a well-written and interesting paper that makes a good contribution to the field of land-ice modeling. I do have a few comments/questions/concerns that I would like the authors to address in a revised manuscript prior to publication. These are summarized below.

1. The authors are correct that there has been little work in performance portability of existing land-ice dycores. One reference that is worth mentioning in this area is the following recent work involving the portability of the Albany Land-Ice first Order Stokes model of (Tezaur et al. 2015) to GPUs and other next-generation architectures using the Kokkos library and programming model:

J. Watkins, I. Tezaur, I. Demeshko. "A study on the performance portability of the finite element assembly process within the Albany land ice solver", E. van Brummelen, A.

Corsini, S. Perotto, G. Rozza, eds. Numerical Methods for Flows: FEF 2017 Selected Contributions, Elsevier, 2019.

This paper does not present a full end-to-end workflow that is portable to GPUs, however; it focuses on the performance portability of only the finite element assembly time, not the linear solve. It is nonetheless worth adding this reference to the bibliography and literature overview.

2. The discretization utilized in FastICE is a finite difference one on a staggered Cartesian grid. In recent years, many production land-ice models have moved to finite element or finite volume discretizations, as these allow you to use unstructured regionally and/or adaptively refined meshes to reduce the total number of dofs in the computation and allow the concentration of computational power where it is needed, which is not possible with structured uniform Cartesian grids. Moreover, w/ structured uniform Cartesian meshes, one ends up with very crude representations of the ice extent and grounding line. I realize that your reason for choosing finite differences was to utilize stencil-based techniques for approximating spatial derivatives in a way that is amenable to the GPU hardware. Is there any hope of extending the scheme to unstructured grids, perhaps using something like DG?

3. When starting your code, did you consider libraries such as Kokkos and RAJA for performance portability over straight-up CUDA? These libraries select the optimal data layout for the hardware used at compile time, thereby making a code portable to multiple architectures, including NVIDIA GPUs. Your current implementation relies on CUDA, which may be problematic if one wishes to run the code on GPUs not from NVIDIA (e.g. AMD GPUs). This may be important in the near future, as there are some planned open science machines coming out soon that are expected not to have NVIDIA GPUs.

4. Pseudo-transient Jacobian-free methods similar in flavor to those proposed here have shown promise for solving the Navier-Stokes equations on GPUs. These methods

work very well until the problem gets too stiff. In this stiff regime, one typically needs to cut the time step substantially, and a preconditioner/matrix is needed, which can be expensive on GPUs. Realistic land ice problems are in general very stiff, and one has a hard time developing good preconditioners even if one has the Jacobian matrix. The numerical examples described in the test case are very simple verification problems. I worry about how the method will perform on realistic problems. It would be good to see one such example in the paper to alleviate this concern. Of particular interest would be a test case with floating ice (e.g. Antarctica simulation), which can pose a lot of challenges for the solver (see R. Tuminaro, M. Perego, I. Tezaur, A. Salinger, S. Price. "A matrix dependent/algebraic multigrid approach for extruded meshes with applications to ice sheet modeling", SIAM J. Sci. Comput. 38(5) (2016) C504-C532). Something simpler to try before doing Antarctica would be a test case with floating ice, e.g. confined shelf, circular shelf.

5. Is CUDA unified virtual memory (UVM) utilized in the implementation, or the memory is managed manually? I assume the latter, but it would be good to state this in the paper. A lot of implementation rely on CUDA UVM, and I think one should move away from that to get the best performance – your paper may make a case for that.

6. The authors introduce the non-dimensionalization of the governing equations as something that is needed for studying the effect of single vs. double precision on the computations (which makes a lot of sense). The study of single vs. double precision arithmetic seems not that rigorous to me, however. Most of the cases were run with double precision, with a couple run single precision, and the authors don't really seem to draw any meaningful conclusions from these results. The effect of reduced/mixed precision arithmetic in continental scale land ice (and more broadly climate) applications is a very interesting research area, which can be formulated as a sensitivity problem and could merit its own publication. I suggest the authors either streamline the single vs. double precision arithmetic discussion, or cut it from this paper, saving it for a later follow on publication where it can be given the proper attention.

7. I am confused about the different resolutions of grids b/w the Elmer/ICE and FastICE computations (e.g. experiments 1 and 2). The codes are quite different as are the techniques therein (e.g. different disrcretizations – PSPG stabilized FEM for Elmer/ICE vs. staggered finite difference for FastICE) so it's hard to say which mesh resolution in Elmer/ICE will be "comparable" to one in FastICE. You must have had some reason for selecting the relative resolutions you considered – can you please explain this here and in the paper? It is difficult to convince the reader that the verification is rigorous w/o explaining discrepancies such as this one.

8. Along the lines of the previous comment, I do not like the discrepancies b/w Elmer/ICE and FastICE for experiment 2. Your theory about the pinning seems plausible, but you should really get to the bottom of this prior to publishing this manuscript.

9. Note that Elmer/ICE uses PSPG stabilization for the full Stokes equations rather than using inf-sup stable velocity-pressure finite elements. This may be worth keeping in mind when making comparisons to Elmer/ICE results.

10. I would be interested to see still more rigorous verification of FastICE, for example, convergence analyses with grid refinement. One can do this on a method of manufactured solutions problem (see

W. Leng, L. Ju, M. Gunzburger, S. Price. "Manufactured solutions and the verification of three-dimensional Stokes ice-sheet models", The Cryosphere 7 19-29, 2013.

for some MMS tests for the full Stokes equations) or by performing a convergence study w.r.t. a reference solution on a fine mesh on a canonical test case: ISMIP-HOM, Dome, Circular Shelf, Confined Shelf, etc. This is important for creating a culture of verification within the climate modeling community, and also to provide evidence that your results are trusted.

11. In my opinion, including the MATLAB and Elmer/ICE results in the computational performance section of the paper is somewhat misleading/confusing, given that the

runs are only on a single core CPU and not representative of CPU hardware capabilities. I am not sure one can make a conclusion from the results that the CPU algorithms are "bad" and the GPU ones are "good". To do a fair comparison you would have to, for instance, take 1 node of a machine with CPUs, max it out, and run Elmer/ICE, then repeat the same procedure for 1 node + GPUs, and look at the relative CPU times. Are you able to perform a study like this? I strongly suggest that you do this and modify the results to have a fair comparison and to avoid misleading the reader.

12. Ultimately, when you get to "real" ice sheet calculations, you will need a thickness solver, to determine how your geometry will change in time. This would need to be coupled with your temperature and velocity equations. Is adding the thickness solver the next step? Please sketch out how that will fit in with your algorithm and maintain performance on GPUs.

13. On p. 29: you state that you "established that a relatively high spatial numerical resolution is necessary to resolve the non-linear and spontaneous localisation of thermomechanically coupled ice flow, including more than 100 grid-points in the vertical direction". Can you please expand on this? It doesn't seem like you really studied the effect of vertical resolution in the problems presented, and this study would be more meaningful on more realistic land ice geometries than those considered. 100 grid points in the vertical dimension would be a lot more than is currently used in practice (most land ice models use on the order of 10 finite elements in the vertical dimension regardless of the horizontal spatial resolution although there is some evidence that more layers may be needed for finer resolution problems in (Tezaur et al. 2015)).

Please address also the following minor comments/typos:

- On p. 1, line 19: you imply that the models in parentheses (Bueler and Brown, 2009; Bassis, 2010; ….) are all shallow ice models, which is not true. For instance, the (Perego et al 2012) and (Tezaur et al. 2015) references are based on the first order Stokes equations, which are derived using a hydrostatic approximation together with

the assumption that the ice sheet is thin. The (Bueler and Brown, 2009) reference focuses on the shallow shelf approximation, not the shallow ice approximation. A simple fix would be to change "such as shallow ice models" to "such as first-order Stokes (refs), shallow shelf (ref) and shallow ice (ref) models".

- P. 2, line 43: since you define CPU, you should also define GPU.

- Title of Section 3 should be "Leveraging".

- Title of Section 5.4: should be "Experiment 4" instead of "Experiment 3".

- P. 29, like 554: "lever" should be "leverage".

―――――――――――――――――

---

## Author Comment (AC1) · 6 Dec 2019

**Referee's comment 1**

The manuscript is well-written, and the governing equations are clearly presented. Overall the manuscript was enjoyable to read, and I learned a lot. The tests were also convincing – as convincing as visual comparisons of results can be. In general, the structure of the manuscript is traditional: Introduce new model, explain the basic principles and the implementation, and test the model against other models. This is fine, and it provides a convenient reference for later work. However, as a reader I would have liked to see a demonstration of what the new model can really do – just

a sneak peek into the suite of problems that the authors hope to address with this new model. There are so many ice models being presented, but it is unfortunately surprisingly rare that we see ice-sheet models applied in ways that make us wiser. So, if possible, I encourage the authors to include a demonstration of the model toward the end of the manuscript – something that is visually, and intellectually, more appealing than the benchmark tests.

**Author's reply 1**

Thank you for your encouraging feedback. We agree that there is an increasing number of ice models and that it is not always clear what the specific contribution of these models to ice dynamics is. The motivation behind developing this model is to develop a process-based model that affords the necessary 3D resolution to capture englacial strain localisation. This process may be of critical importance in the boundaries of fast flow like the basal interface, grounding zones and shear margins. It is also a subtle component of the overall ice dynamics and requires a careful assessment of when and why it becomes relevant and which locations on our ice sheets might serve as test sites for the model predictions. We are currently working on two follow-up manuscripts applying this code to the flow-to-sliding transition and to shear margin stability. As you mention, developing sophisticated models and advancing our understanding of ice dynamics are two distinct challenges. The first is a necessary but not a sufficient condition for the latter. To do justice to both, we prefer to focus on the numerical methods, benchmarking and performance evaluation for this manuscript and leverage this code for advancing our understanding of ice dynamics in a separate manuscript that we will submit to a glaciological journal. While we agree that an actual application case is more appealing and interesting than benchmarks, we believe that this code can help us make progress on important, fundamental questions in glaciology and we prefer to develop this potential fully in our separate contributions. We are happy to make preliminary results available to you to demonstrate the value of the code for these problem. For this manuscript, we have included a more detailed motivation for

this kind of code and more extensive reference to the problems for which it is relevant.

**Referee's comment 2**

Line 36: The GPU-acceleration is very interesting and, as far as I know, rather new in ice-sheet models. However, a quick search leads to Brædstrup et al. (2014) "Ice-sheet modelling accelerated by graphic cards" in Computers & Geosciences 72, 210-220. This paper is not cited here, although it covers some of the same challenges and principles of GPU-acceleration.

**Author's reply 2**

Thank you for pointing this out. We indeed overlooked the citation of the work from Brædstrup et al. (2014).

**Changes in the manuscript 2**

We added reference to this work in the revised manuscript at line 61: "We tailor our numerical method to optimally exploit the massive parallelism of GPU hardware, taking inspiration from recent successful GPU-based implementations of viscous and coupled flow problems (Brædstrup et al., 2014; Omlin, 2017; Räss et al., 2018; Duretz et al., 2019; Räss et al., 2019a)."

**Referee's comment 3**

line 42: Also, regarding GPU-acceleration, it would be good to see reference to other flow problems that have successfully been GPU accelerated. What problems and models have inspired the authors?

**Author's reply 3**

We rephrased in a more explicit way the source of inspiration of the GPU-based FastICE implementation (line 61).

**Changes in the manuscript 3**

[Figure]

Line 61: "We tailor our numerical method to optimally exploit the massive parallelism of GPU hardware, taking inspiration from recent successful GPU-based implementations of viscous and coupled flow problems (Brædstrup et al., 2014; Omlin, 2017; Räss et al., 2018; Duretz et al., 2019; Räss et al., 2019a)."

**Referee's comment 4**

line 122: The comment on single-precision calculations leaves me confused. Are the GPU-calculations single precision? Or does it depend on the specific GPU architecture? Please clarify.

**Author's reply 4**

The benchmarks and calculations in this study are performed using double precision arithmetic if not specified otherwise. We reported single precision efficiency to show the potential performance gain from reducing the arithmetic precision of the calculations. Until recently, it was commonly admitted and implicitly assumed that scientific calculations are (and should be) performed using double precision floating point arithmetic. This choice goes back a couple of decades ago when hardware was computation-bounded; double precision would provide enhanced convergence, thus more efficient calculations, since less floating operations were needed. However, we nowadays observe a shift towards memory-bounded hardware and software where transferring memory (numbers) is more limiting compared to performing arithmetic operations. Thus single or half precision calculation may become interesting as the numbers take twice or four time less amount of memory - which results in factor 2 or 4 performance increase. Alternatively, similar performance can be observed for a two or four-times increase in the numerical grid resolution. Future work may address whether performing calculations using lower arithmetic precision but increased numerical grid resolutions can outperform well-established double precision calculations. A detailed assessment of the issue may deserve separate publication.

**Changes in the manuscript 4**
Line 257: "The computations in CUDA C shown in the remainder of the paper were performed using double-precision arithmetic, if not specified otherwise."

**Referee's comment 5**

line 163: Braedstrup et al has a nice description of staggered grids and GPU acceleration – must be cited here.

**Author's reply 5**

Although we do not question the accurate description of the staggered grid from Braestrup et al., they use a Gauss-Seidel solver in their study, which shows some limitations in terms of parallel implementation. The solve they use requires information from neighbouring cells at each iteration which may, when executed in parallel, lead to read/write conflicts. Our PT solver relies on a fully parallel iteration strategy, which inherently takes care of updating the entire field of old values with updated ones thus circumventing the neighbouring cell read/write issues and avoiding to rely on a "red-black" type of scheme. We are now citing the suggested work, just not with specific reference to the staggered grid setup.

**Changes in the manuscript 5**

–

**Referee's comment 6**

line 175: Even up-wind advection schemes are going to suffer from numerical diffusion – and high numerical resolution is just making it worse. Please discuss this here.

**Author's reply 6**

True, upwind scheme also suffer from numerical diffusion. To ensure that our numerical results are not confounded by numerical diffusion, we set the numerical resolution such that the Grid Peclet number is smaller than the physical Peclet number, i.e. $n\_x > L\_x * v\_x / 2$. Limiting numerical diffusion is one motivation for using high numerical

resolution in our computations.

**Changes in the manuscript 6**

We have added the following clarification to the paragraph on line 183: "To ensure that our numerical results are not confounded by numerical diffusion, the Grid Peclet number must be smaller than the physical Peclet number. Limiting numerical diffusion is one motivation for using high numerical resolution in our computations."

**Referee's comment 7**

line 182: The matrix-free solver using pseudo-time is nicely explained. However, it would be good to see exactly how the residuals propagate in the grid. Many similar matrix-free relaxation schemes use multi-grid setups to make the residuals decay faster – these could be discussed.

**Author's reply 7**

An excellent point, thanks for bringing it up. We have included an additional figure in section 5.5 displaying the decay of the residual as function of the damping parameter. Multi-Grid configuration are an alternative solution improving residual decay. However, MG methods may generate quite some overhead by the addition of multiple grid levels and may hinder performance by restriction and prolongation operators. Also, coarser grid may not saturate the GPU and result in a drop of efficiency.

**Changes in the manuscript 7**

Line 247: "The iteration count increases with the numerical problem size for second-order PT solvers scales close to ideal multi-grid implementations. However, the main advantage of the PT approach is its conciseness and the fact that only one additional read/write operation needs to be included - keeping additional memory transfers to the strict minimum."

**Referee's comment 8**

Eqn. 15: I believe that theta < 1 is often referred to as under-relaxation.

**Author's reply 8**

The variable is a scalar we use to select the fraction of a given nonlinear quantity to be updated each iteration. When theta=0, we would always use the initial guess, while theta=1, we would take 100% of the current nonlinear quantity. We usually define theta to be in the range of 1e-2 - 1e-1 in order to account for some time to fully relax the nonlinear quantities as the nonlinear problem may not be sufficiently converged at the beginning of the iterations. This approach is in a way similar to an under-relaxation scheme.

**Changes in the manuscript 8**

Line 204-209: "We use the scalar [. . .] to select the fraction of a given nonlinear quantity, here the effective viscosity [. . .], to be updated each iteration. When =0, we would always use the initial guess, while =1, we would take 100% of the current nonlinear quantity. We usually define theta to be in the range of [. . .] in order to account for some time to fully relax the nonlinear viscosity as the nonlinear problem may not be sufficiently converged at the beginning of the iterations. This approach is in a way similar to an under-relaxation scheme and was successfully implemented in the ice sheet model development by Tezaur (2015), for example."

**Referee's comment 9**

Eqn. 19: Again, I miss information on how the residuals decay in the grid – particularly when using this stabilizing scheme. Also, I could not find previous reference to alpha, but I might have missed it.

**Author's reply 9**

Thank you for pointing out the missing alpha definition. We no longer use alpha in the manuscript, replacing it explicitly for enhanced clarity in Eqn. 19. We have also added a Figure 16 in the new Section 5.5 displaying a) the residuals' convergence

history for a 2-D simulation and b) the impact of the "stabilising" scheme as function of the damping parameter nu in terms of the total number of iteration count to reach convergence threshold.

**Changes in the manuscript 9**

Line 453-458 and Figure 16.

**Referee's comment 10**

line 299: I can see how the non-dimensional equation makes the implementation simpler, but is it necessary to present results in the non-dimensional form? It just makes the output harder to understand.

**Author's reply 10**

As you point out, presenting results in a non-dimensional form has advantages and drawbacks. Dimensional results are more intuitive and easier to compare to observations, but non-dimensional results are more general and can be scaled back easily using the scales provided in Eqns 7 and 9 to various configurations without having to re-run the model. Here, we prefer the generality of non-dimensionality since we are looking at generic benchmark cases instead of applying our model to a particular field site or comparing against specific field measurements.

**Changes in the manuscript 10**

–

**Referee's comment 11**

Section 5: There is some repetition of captions in the text. "In Figure 4, we plot. . ."; "Figure 5 shows. . ."; "Figure 6 shows. . ." etc. This could be skipped to make the text smoother.

**Author's reply 11**

Thank you for pointing this out.

**Changes in the manuscript 11**

We re-phrased Section 5.1 and 5.2 avoiding the figure caption repetition for better clarity. Please refer to the revised text in Section 5 for updates.

**Referee's comment 12**

line 308: Why are the benchmark tests performed at different resolutions? Does the GPU-model require order-of-magnitude more DOFs to yield the same accuracy as the FEM model? The comparisons give leave me with that impression, and then what is the advantage of the PT setup?

**Author's reply 12**

Thank you for raising this important point. The benchmark tests where originally run at higher resolutions with the FastICE GPU code since we can afford it. The Elmer/Ice results are obtained on the largest available single-core/direct solver resolution (or robust iterative solver for the 3D case). The latest results for the benchmark of experiment 2 show the good agreement among FastICE and Elmer/Ice at comparable resolutions. However, discrepancy between low and high numerical grid resolutions suggest that although the two different solution strategies match, they both may not fully capture the physics with accuracy at low resolutions in some cases, such as the 3D benchmark of Experiment 2. We report this issue in a new Figure 15 in the Section 5.5, showing the convergence of the numerical implementation among grid refinement.

**Changes in the manuscript 12**

Lines 441-458: We added a new Section "5.5: Validation of the FastICE numerical implementation" to discuss this topic and a related Figure 15.

**Referee's comment 13 line 314: "numerical resolution grid resolution"**

**Author's reply 13**

Thank you for pointing this out. We corrected the sentence. Which now reads:

**Changes in the manuscript 13**

Line 329: ". . .and used a numerical grid resolution. . ."

**Referee's comment 14**

line 333: The authors are right to address the discrepancies between the model results – but why not follow up on the idea to pin nodes in the FEM mesh?

**Author's reply 14**

We re-evaluated the benchmark test case using a comparable numerical grid resolution for our FastICE GPU solver and for Elmer/Ice. The result now agree for a particular numerical grid resolution. However, discrepancy with previous results suggest that the numerical resolution used to compare the two software may not be sufficient to resolve the physical process. To address this second limitation, we provide one additional figure showing the convergence of our method with and increase in numerical grid resolution and comparing the results to a high-resolution "reference" simulation.

**Changes in the manuscript 14**

We updated the Figure 7 with the latest benchmark test results at similar numerical grid resolutions between FastICE and Elmer/Ice and adapted the text from Section 5.2. Lines 441-458: We added a new Section "5.5: Validation of the FastICE numerical implementation" to discuss this topic and a related new Figure 15.

**Referee's comment 15**

Fig. 15: The performance diagrams are very convincing – however, the use of widely different DOFs for the FEM and PT models in the benchmark tests makes we wonder if the speedup is real?

**Author's reply 15**

The purpose of these graphs is not to report speed-up versus single-core Matlab or Elmer/ice, but to inform the reader about the potential and the scaling of the iterative and matrix-free PT approach to handle large number of grid points representative of high-resolutions simulations. In terms of high-performance "desktop" computing - what certainly majority of the researcher still rely on - it is fair to compare the range of affordable DOF for the FEM and PT implementations. Finally, high resolution calculations affordable with the PT approach may become necessary when resolving internal deformation localising into self-consistent formation of boundary layers prone to a sliding-like behaviour.

**Changes in the manuscript 15**

–

Sincerely yours,

Ludovic Räss, on behalf of the authors.

––––––––––––––––––––––––––

---

## Author Comment (AC2) · 6 Dec 2019

**Referee's comment 1**

The authors are correct that there has been little work in performance portability of existing land-ice dycores. One reference that is worth mentioning in this area is the following recent work involving the portability of the Albany Land-Ice first Order Stokes model of (Tezaur et al. 2015) to GPUs and other next-generation architectures using the Kokkos library and programming model: J. Watkins, I. Tezaur, I. Demeshko. "A study on the performance portability of the finite element assembly process within the Albany land ice solver", E. van Brummelen, A. Corsini, S. Perotto, G. Rozza, eds.

Numerical Methods for Flows: FEF 2017 Selected Contributions, Elsevier, 2019. This paper does not present a full end-to-end workflow that is portable to GPUs, however; it focuses on the performance portability of only the finite element assembly time, not the linear solve. It is nonetheless worth adding this reference to the bibliography and literature overview.

**Author's reply 1**

Thank you for suggesting this reference on related topics. We have included it into our manuscript.

**Changes in the manuscript 1**

Line 63: "Our work contributes to the few land-ice dynamical cores targeting many-cores architectures such as GPUs (Brædstrup et al., 2014; Watkins et al., 2019)"

**Referee's comment 2**

The discretization utilized in FastICE is a finite difference one on a staggered Cartesian grid. In recent years, many production land-ice models have moved to finite element or finite volume discretisations, as these allow you to use unstructured regionally and/or adaptively refined meshes to reduce the total number of dofs in the computation and allow the concentration of computational power where it is needed, which is not possible with structured uniform Cartesian grids. Moreover, w/ structured uniform Cartesian meshes, one ends up with very crude representations of the ice extent and grounding line. I realize that your reason for choosing finite differences was to utilize stencil-based techniques for approximating spatial derivatives in a way that is amenable to the GPU hardware. Is there any hope of extending the scheme to unstructured grids, perhaps using something like DG?

**Author's reply 2**

Indeed, many large-scale ice models have moved to finite elements to conform to complex basal topography and other geometric complexities arising in the grounding zone

or on ice shelfs. The motivation behind FastICE is develop a complementary tool to existing approaches that enables us to better model and understand englacial instabilities such as thermo-mechanical localisation at the scale of individual field sites. Thermo-mechanical localisation arise in a self-consistent way in shear margins, at the grounding zone or in the vicinity of the basal sliding interface, but the degree and location of localisation is not known apriori. A body-fitted mesh is hence less valuable for our purposes than for problems with fixed geometry. Grid adaptivity could be beneficial and we have used it in previous problems that were dominated by singularities (e.g., Suckale et al., 2014). Recent work, however, suggests that singularities are blunted dynamically and that the flow field exhibits significant 3D variability throughout the entire boundary layer. The goal of FastICE is to better understand the physical processes governing this small-scale variability by quantifying the observational signature of different processes and comparing these model predictions against observational data at the field-site, rather than the regional, scale. You are of course correct in pointing out that Cartesian uniform meshes combined with the Finite-difference method enable the numerical application to run in parallel on GPUs close to hardware limit, but amenability of our grid setup to the GPU hardware is only one reason for opting for a Cartesian grid. The more important difference is that FastICE is targeting other scientific problems than many existing land-ice models. We added it t the discussion.

**Changes in the manuscript 2**

Line 539-543: "To address these limitations, we have developed FastICE, a new parallel GPU-based numerical model. The goal of FastICEis to better understand the physical processes that govern englacial instabilities such as thermomechanical localisation at the field-site, rather than the regional, scale. It hence targets other scientific problems than many existing land-ice models and complements these previous models."

**Referee's comment 3**

When starting your code, did you consider libraries such as Kokkos and RAJA for performance portability over straight-up CUDA? These libraries select the optimal data layout for the hardware used at compile time, thereby making a code portable to multiple architectures, including NVIDIA GPUs. Your current implementation relies on CUDA, which may be problematic if one wishes to run the code on GPUs not from NVIDIA (e.g. AMD GPUs). This may be important in the near future, as there are some planned open science machines coming out soon that are expected not to have NVIDIA GPUs.

**Author's reply 3**

Code portability is an important point, thank you for raising it. FastICE development aligns within a general effort to spread high-performance, parallel and super computing to Earth sciences. Usually performance and portability are rather opposite as a general and portable implementation may trade off performance, and vice-versa. However, the vectorised CUDA indexes could be replaced by explicit loops that can be parallelised using a shared memory approach (such e.g. openMP). Regarding various GPU designs, there are active development efforts by the broader community of wrappers to enable porting CUDA-based code to AMD or Intel GPUs.

**Changes in the manuscript 3**

–

**Referee's comment 4**

Pseudo-transient Jacobian-free methods similar in flavor to those proposed here have shown promise for solving the Navier-Stokes equations on GPUs. These methods work very well until the problem gets too stiff. In this stiff regime, one typically needs to cut the time step substantially, and a preconditioner/matrix is needed, which can be expensive on GPUs. Realistic land ice problems are in general very stiff, and one has a hard time developing good preconditioners even if one has the Jacobian matrix. The numerical examples described in the test case are very simple verification problems.

I worry about how the method will perform on realistic problems. It would be good to see one such example in the paper to alleviate this concern. Of particular interest would be a test case with floating ice (e.g. Antarctica simulation), which can pose a lot of challenges for the solver (see R. Tuminaro, M. Perego, I. Tezaur, A. Salinger, S. Price. "A matrix dependent/algebraic multigrid approach for extruded meshes with applications to ice sheet modeling", SIAM J. Sci. Comput. 38(5) (2016) C504-C532). Something simpler to try before doing Antarctica would be a test case with floating ice, e.g. confined shelf, circular shelf.

**Author's reply 4**

An important point, thank you for raising it. Stiffness is indeed a concern in ice-sheet modelling, but it is a challenge not only for numerical reasons. Rather, it is a reflection of changing physical processes that govern ice flow at different scales and also at different locations along outlet glaciers and ice streams. One approach to tackling that challenge is to focus on numerical techniques suited specifically for stiff problems. Another is to focus on understanding the physical processes that lead to stiff behaviour in the first place and adjust the governing equations in suitable ways to represent these. The philosophy behind FastICE is the latter approach. We argue that specific locations on ice sheets like shear margins, grounding zones and the basal sliding interface require a multi-physics approach that could be built into FastICE. You mention the example of ice shelfs, which is of course at the heart of the current debate about sea-level-rise projections. There are many challenges in better understanding the coupling between ice shelfs, the ocean, and land ice including the ice-cliff instability (which requires a brittle rheology and failure model), the vulnerability of ice shelfs to meltwater ponding at the surface (which requires an englacial hydrology model), and the dynamics of the grounding zone (which requires a free-boundary model). Needless to say, ultimately we need both, better numerical techniques for stiff problems and a better physical understanding. Since we focus primarily on the field-site rather than the regional or ice-sheet scale, some of the large-scale numerical issues like stiffness are

less of a problem for the applications that we are interested in. We clarified the motivations behind FastICE and how our model complements existing approaches rather than attempting to replace them.

**Changes in the manuscript 4**

Line 245-256: "Many large-scale ice models have moved to finite elements to conform to complex basal topography and other geometric complexities arising in the grounding zone or on ice shelves. The motivation behind FastICE is develop a complementary tool to existing approaches that enables us to better model and understand englacial instabilities such as thermomechanical localisation at the scale of individual field sites. Thermomechanical localisation arises in a self-consistent way in shear margins, at the grounding zone or in the vicinity of the basal sliding interface, but the degree and location of localisation is not known apriori. A body-fitted mesh is hence less valuable for our purposes than for problems with fixed geometry. Grid adaptivity could be beneficial and we have used it in previous problems that were dominated by singularities [. . .]. Recent work, however, suggests that singularities are blunted dynamically and that the flow field exhibits significant 3-D variability throughout the entire boundary layer. The goal of FastICE is to better understand the physical processes governing this small-scale variability by quantifying the observational signature of different processes and comparing these model predictions against observational data at the field-site, rather than the regional, scale. FastICE is targeting other scientific problems than many existing land-ice models."

**Referee's comment 5**

Is CUDA unified virtual memory (UVM) utilized in the implementation, or the memory is managed manually? I assume the latter, but it would be good to state this in the paper. A lot of implementation rely on CUDA UVM, and I think one should move away from that to get the best performance – your paper may make a case for that.

**Author's reply 5**

Thank you for pointing out the need to clarify memory management. Our implementation does indeed not rely on the UVM features from CUDA, because at the time we initiated the work and later on assessed the UVM performance (early 2018), UVM was showing about one order of magnitude lower performance. We suspect the internal memory handling to be responsible of constantly synchronising host and device memory, which is not needed in our case. We clarified this by adding a statement in the Section 3.1.

**Changes in the manuscript 5**

Line 273: "Our implementation does not rely on the CUDA unified virtual memory (UVM) features. UVM avoids to explicitly define data transfer between the host (CPU) and device (GPU) arrays but results in about one order of magnitude lower performance. We suspect the internal memory handling to be responsible of continuously synchronising host and device memory, which is not needed in our case."

**Referee's comment 6**

The authors introduce the non-dimensionalization of the governing equations as something that is needed for studying the effect of single vs. double precision on the computations (which makes a lot of sense). The study of single vs. double precision arithmetic seems not that rigorous to me, however. Most of the cases were run with double precision, with a couple run single precision, and the authors don't really seem to draw any meaningful conclusions from these results. The effect of reduced/mixed precision arithmetic in continental scale land ice (and more broadly climate) applications is a very interesting research area, which can be formulated as a sensitivity problem and could merit its own publication. I suggest the authors either streamline the single vs. double precision arithmetic discussion, or cut it from this paper, saving it for a later follow on publication where it can be given the proper attention.

**Author's reply 6**

The choice of arithmetic precision is an important topic and merits an in-depth assessment resulting its own publication (see also response 4 to review #1). Our current study does not aim at investigating the effects, benefits and drawbacks of various arithmetic precision implementations. Although not in the current spotlight, we still wish to highlight the ability of our model to perform using single precision floating point arithmetics. Together with the non-dimensional for of the governing equations, the features pave the path for future studies addressing these important issues related to lower precision arithmetic and their benefits in light of memory bounded applications.

**Changes in the manuscript 6**

Line 257: "The computations in CUDA C shown in the remainder of the paper were performed using double-precision arithmetic, if not specified otherwise."

**Referee's comment 7**

I am confused about the different resolutions of grids b/w the Elmer/ICE and FastICE computations (e.g. experiments 1 and 2). The codes are quite different as are the techniques therein (e.g. different disrcretizations – PSPG stabilized FEM for Elmer/ICE vs. staggered finite difference for FastICE) so it's hard to say which mesh resolution in Elmer/ICE will be "comparable" to one in FastICE. You must have had some reason for selecting the relative resolutions you considered – can you please explain this here and in the paper? It is difficult to convince the reader that the verification is rigorous w/o explaining discrepancies such as this one.

**Author's reply 7**

You are correct pointing out it is hard to say what are the optimal mesh resolutions in order to compare various discretisation and numerical methods. For the benchmark, we decided to employ as large as possible numerical resolutions that would still deliver results in "reasonable" (day-scale) wall-times while running on desktop-type of computer hardware (single CPU - single GPU). For optimal comparison, we selected rectangu-

lar mesh elements within the Elmer/Ice FEM framework; we are confident about our choice to be a reasonable comparison involving similar regular spatial discretisation. The two solving approaches should deliver similar results independently of the numerical implementations. We addressed this in the result section.

**Changes in the manuscript 7**

Line 329-334: "We use higher numerical grid resolution within FastICE as we can afford it. Varying the numerical resolution also permits to test both the agreement between to different numerical approaches and convergence. The fact that we obtain matching results when increasing grid resolution significantly suggests that we resolve the relevant physical processes sufficiently, even at lower resolutions. We report an exception to this trend in the 3-D case of Experiment 2."

**Referee's comment 8**

Along the lines of the previous comment, I do not like the discrepancies b/w Elmer/ICE and FastICE for experiment 2. Your theory about the pinning seems plausible, but you should really get to the bottom of this prior to publishing this manuscript.

**Author's reply 8**

We addresses the issue regarding the discrepancy between FastICE and Elmer/Ice in the 3D configuration of experiment 2. We repeated the benchmark using similar gird resolution in FastICE than Elmer/Ice and the results agree. We are thus confident FastICE reproduces the benchmark tests with similar accuracy than Elmer does. However, our original results suggests that the spatial resolution at which the benchmark is performed may not be sufficient in order to achieve convergence of the numerical results. We investigated this issue by performing an additional test refining the numerical grid resolution from coarse to a reference numerical solution on a fine grid. We show convergence of the method among grid refinement.

**Changes in the manuscript 8**

Lines 441-458: We added a new Section "5.5: Validation of the FastICE numerical implementation" to discuss this topic and a related new Figure 15.

**Referee's comment 9**

Note that Elmer/ICE uses PSPG stabilization for the full Stokes equations rather than using inf-sup stable velocity-pressure finite elements. This may be worth keeping in mind when making comparisons to Elmer/ICE results. # Author's reply 9

Yes, thank you for pointing this out.

**Changes in the manuscript 9**

–

**Referee's comment 10**

I would be interested to see still more rigorous verification of FastICE, for example, convergence analyses with grid refinement. One can do this on a method of manufactured solutions problem (see W. Leng, L. Ju, M. Gunzburger, S. Price. "Manufactured solutions and the verification of three-dimensional Stokes ice-sheet models", The Cryosphere 7 19-29, 2013. for some MMS tests for the full Stokes equations) or by performing a convergence study w.r.t. a reference solution on a fine mesh on a canonical test case: ISMIP-HOM, Dome, Circular Shelf, Confined Shelf, etc. This is important for creating a culture of verification within the climate modeling community, and also to provide evidence that your results are trusted.

**Author's reply 10**

We agree and support the importance of a culture of verification within the climate modelling community (and beyond). We thus provided an additional figure reporting the convergence of our method for a given configuration among increase of the numerical grid resolution. We report that our method is first order accurate (expected from the finite-difference approximation) with regards to high-resolution reference results in both

2-D and 3-D.

**Changes in the manuscript 10**

Lines 441-458: We added a new Section "5.5: Validation of the FastICE numerical implementation" to discuss this topic and a related new Figure 16.

**Referee's comment 11**

In my opinion, including the MATLAB and Elmer/ICE results in the computational performance section of the paper is somewhat misleading/confusing, given that the runs are only on a single core CPU and not representative of CPU hardware capabilities. I am not sure one can make a conclusion from the results that the CPU algorithms are "bad" and the GPU ones are "good". To do a fair comparison you would have to, for instance, take 1 node of a machine with CPUs, max it out, and run Elmer/ICE, then repeat the same procedure for 1 node + GPUs, and look at the relative CPU times. Are you able to perform a study like this? I strongly suggest that you do this and modify the results to have a fair comparison and to avoid misleading the reader.

**Author's reply 11**

We support your comment and agree one should not jump to conclusions about an algorithm being "bad" or "good" based on those single-core CPU results displayed besides GPU-based results. However, those are just facts and we want to show what value to expect in our metric for a single-core CPU process. Due to the infinite number of possible node configurations, I do not think that one could ever make a relevant comparison. This motivated our choice to report the following results. We compared non MPI Elmer/Ice runtime on a desktop machine versus a non MPI FastICE runtime on a single desktop GPU, with the drawback that CPU utilisation is not maximised by construction while GPU utilisation is. Finally, we are mostly interested to report the scaling of the fastICE runtime with increase in problem size rather than to perform and extensive comparison among FastICE and Elmer/Ice as performance cannot be fairly

compared given the different approaches.

**Changes in the manuscript 11**

–

**Referee's comment 12**

Ultimately, when you get to "real" ice sheet calculations, you will need a thickness solver, to determine how your geometry will change in time. This would need to be coupled with your temperature and velocity equations. Is adding the thickness solver the next step? Please sketch out how that will fit in with your algorithm and maintain performance on GPUs.

**Author's reply 12**

Indeed, including a thickness solver could be one way forward. That being said, our primary goal with FastICE is an improved process-based understanding of the boundaries of fast flow including shear margins, grounding zones and the basal sliding interface instead of focusing on "real" ice-sheet calculations for which several models already exist. Recent studies (e.g., Elsworth and Suckale, 2016) have shown that shear margin locations can shift almost discontinuously over as little as a few months if their location is governed by subglacial hydrology. These rapid adjustments of the sliding interface are an important contributor to the uncertainty in near-term sea-level-rise projections and are currently our primary focus. In most locations, with the possible exception of Thwaites Glacier, ice thickness will change very little on the monthly to annual time scale. With that scope in mind, a thickness solver is less important than integrating multi-physics behavior such as englacial and subglacial hydrology. There is no general answer on how these multi-physics components will alter GPU performance and we agree that a careful implementation is necessary to maintain scalability. That being said, the pseudo-transient algorithm behind FastICE lends itself to the integration of other components and can be tailored to the need of future specific studies.

**Changes in the manuscript 12**

–

**Referee's comment 13**

On p. 29: you state that you "established that a relatively high spatial numerical resolution is necessary to resolve the non-linear and spontaneous localisation of thermomechanically coupled ice flow, including more than 100 grid-points in the vertical direction". Can you please expand on this? It doesn't seem like you really studied the effect of vertical resolution in the problems presented, and this study would be more meaningful on more realistic land ice geometries than those considered. 100 grid points in the vertical dimension would be a lot more than is currently used in practice (most land ice models use on the order of 10 finite elements in the vertical dimension regardless of the horizontal spatial resolution although there is some evidence that more layers may be needed for finer resolution problems in (Tezaur et al. 2015)).

**Author's reply 13**

High vertical (and horizontal) resolution will be needed to resolve local stress and pressure gradient arising from interaction with non-flat topography or to dynamically capture the localisation of strain and heat in the formation of shear-zones such as internal sliding layers (see attached figure). Those results are in consideration for publication in a separate study.

**Changes in the manuscript 13**

–

**Referee's comment 14**

Please address also the following minor comments/typos:

p. 1, line 19: you imply that the models in parentheses (Bueler and Brown, 2009; Bassis, 2010; ....)  are all shallow ice models, which is not true.  For instance, the

(Perego et al 2012) and (Tezaur et al. 2015) references are based on the first order Stokes equations, which are derived using a hydrostatic approximation together with the assumption that the ice sheet is thin. The (Bueler and Brown, 2009) reference focuses on the shallow shelf approximation, not the shallow ice approximation. A simple fix would be to change "such as shallow ice models" to "such as first-order Stokes (refs), shallow shelf (ref) and shallow ice (ref) models".

P. 2, line 43: since you define CPU, you should also define GPU.

Title of Section 3 should be "Leveraging".

Title of Section 5.4: should be "Experiment 4" instead of "Experiment 3".

P. 29, like 554: "lever" should be "leverage".

**Author's reply 14**

Thank you for your suggestions. We rephrased that portion of the introduction following your guideline:

GPU is defined 6 lines previous to the definition of CPU.

To lever (verb), to lever + age (noun), So the verb is to lever and not to leverage (see this link https://this.isfluent.com/blog/2010/are-you-stupid-enough-to-use-leverage-as-a-verb for further details - apologies for the somewhat inappropriate language).

Experiment 4 is a variation of Experiment 3. We thus renamed them Experiments 3a and 3b for enhanced readability.

**Changes in the manuscript 14**

Please see previous lines.

Sincerely yours,

Ludovic Räss, on behalf of the authors.

[Figure]

[Figure]

Fig. 1.

[Figure]

---

## Author Response (AR2)

Dear Dr. Robel,

Thank you for handling our manuscript and for accepting it for publication in GMD upon addressing the few minor technical revisions.

We took care of editing your suggestions in the final version of the manuscript.

As the first author moved from Stanford to ETH Zurich, would it be possible to include a foot note for Ludovic Räss stating:

Now at:
Laboratory of Hydraulics, Hydrology and Glaciology (VAW), ETH Zurich, Zurich, Switzerland
Swiss Federal Institute for Forest, Snow and Landscape Research (WSL), Birmensdorf, Switzerland

Sincerely yours,

Ludovic Räss, on behalf of the authors.

**Referee's comment 1**

Thank you for revising the manuscripts and addressing my questions/comments. I am satisfied with the revision and your response. I have one minor correction, namely to change "validation" to "verification" in the title of Section 5.5. The results you are presenting in this section are verification results (you are checking that your implementation agrees with theory by checking things like convergence rates), not validation results (validation would be comparing your model output with observational data).

**Authors' reply 1**

Thank you for your feedback regarding the revision. We agree with your latest suggestion and changed "validation" to "verification" in the the title of Section 5.5.

[revised manuscript text omitted]